# Blending Imitation and Reinforcement Learning for Robust Policy Improvement

**Xuefeng Liu**[1]*, **Takuma Yoneda**[2], **Rick L. Stevens**[1], **Matthew R. Walter**[2], **Yuxin Chen**[1]
[1]Department of Computer Science, University of Chicago
[2]Toyota Technological Institute at Chicago

## Abstract

While reinforcement learning (RL) has shown promising performance, its sample complexity continues to be a substantial hurdle, restricting its broader application across a variety of domains. Imitation learning (IL) utilizes oracles to improve sample efficiency, yet it is often constrained by the quality of the oracles deployed. To address the demand for robust policy improvement in real-world scenarios, we introduce a novel algorithm, Robust Policy Improvement (RPI), which actively interleaves between IL and RL based on an online estimate of their performance. RPI draws on the strengths of IL, using oracle queries to facilitate exploration—an aspect that is notably challenging in sparse-reward RL—particularly during the early stages of learning. As learning unfolds, RPI gradually transitions to RL, effectively treating the learned policy as an improved oracle. This algorithm is capable of learning from and improving upon a diverse set of black-box oracles. Integral to RPI are Robust Active Policy Selection (RAPS) and Robust Policy Gradient (RPG), both of which reason over whether to perform state-wise imitation from the oracles or learn from its own value function when the learner's performance surpasses that of the oracles in a specific state. Empirical evaluations and theoretical analysis validate that RPI excels in comparison to existing state-of-the-art methods, showing superior performance across various domains. Please checkout our website[1].

## 1 Introduction

Reinforcement learning (RL) has shown significant advancements, surpassing human capabilities in diverse domains such as Go (Silver et al., 2017), video games (Berner et al., 2019; Mnih et al., 2013), and Poker (Zhao et al., 2022). Despite such achievements, the application of RL is largely constrained by its substantially high sample complexity, particularly in fields like robotics (Singh et al., 2022) and healthcare (Han et al., 2023), where the extensive online interaction for trial and error is often impractical.

Imitation learning (IL) (Osa et al., 2018) improves sample efficiency by allowing the agent to replace some or all environment interactions with demonstrations provided by an oracle policy. The efficacy of IL heavily relies on access to near-optimal oracles for approaches like behavior cloning (Pomerleau, 1988; Zhang et al., 2018) or inverse reinforcement learning (Abbeel and Ng, 2004; Finn et al., 2016; Ho and Ermon, 2016; Ziebart et al., 2008). Interactive IL techniques, such as DAgger (Ross et al., 2011) and AggreVate(D) (Ross and Bagnell, 2014; Sun et al., 2017), similarly assume that the policy we train (i.e., *learner* policy) can obtain demonstrations from a near-optimal oracle. When we have access to rewards, the learner has the potential to improve and outperform the oracle. THOR (Sun et al., 2018) exemplifies this capability by utilizing a near-optimal oracle for cost shaping, optimizing the $k$-step advantage relative to the oracle's value function (referred to as "cost-to-go oracle").

However, in realistic settings, obtaining optimal or near-optimal oracles is often infeasible. Typically, learners have access to *suboptimal* and *black-box* oracles that may not offer optimal trajectories or quantitative performance measures in varying states, requiring substantial environment interactions

---

*Correspondence to: Xuefeng Liu <xuefeng@uchicago.edu>.
[1]https://robust-policy-improvement.github.io/

to identify state-wise optimality. Recent approaches, including LOKI (Cheng et al., 2018) and TGRL (Shenfeld et al., 2023) aim to tackle this issue by combining IL and RL. They focus on a single-oracle setting, whereas MAMBA (Cheng et al., 2020) and MAPS (Liu et al., 2023) learn from multiple oracles. These approaches demonstrate some success, but often operate under the assumption that at least one oracle provides optimal actions in any given state, which does not always hold in practice. In situations where no oracle offers beneficial advice for a specific state, it is more effective to learn based on direct reward feedback. Our work intend to bridge this gap by adaptively blending IL and RL in a unified framework.

**Our contributions.** In this paper, we present max$^+$, a learning framework devised to enable robust learning in unknown Markov decision processes (MDP) by interleaving RL and IL, leveraging multiple suboptimal, black-box oracles. Within this framework, we introduce *Robust Policy Improvement* (RPI), a novel policy gradient algorithm designed to facilitate learning from a set of black-box oracles. RPI comprises two innovative components:

1. *Robust Active Policy Selection* (RAPS), improving value function estimators of black-box oracles efficiently, and
2. *Robust Policy Gradient* (RPG), executing policy gradient updates within an actor-critic framework based on a newly devised advantage function.

Our algorithm strikes a balance between learning from these suboptimal oracles and self improvement through active exploration in states where the learner has surpassed the oracle's performance. We provide a theoretical analysis of our proposed method, proving that it ensures a performance lower bound no worse than that of the competing baseline (Cheng et al., 2020). Through extensive empirical evaluations on eight different tasks from DeepMind Control Suite (Tassa et al., 2018) and Meta-World (Yu et al., 2020), we empirically demonstrate that RPI outperforms contemporary methods and then ablate its core components.

## 2 RELATED WORK

**Online selection of suboptimal experts.** CAMS (Liu et al., 2022b;a) learns from multiple suboptimal black-box experts to perform model selection based on a given context, but is only applicable in stateless online learning settings. Meanwhile, SAC-X (Riedmiller et al., 2018) learns the intention policies (oracles), each of which optimizes their own auxiliary reward function, and then reasons over which of these oracles to execute as a form of curriculum learning for the task policy. LfGP (Ablett et al., 2023) combines adversarial IL with SAC-X to improve exploration. Defining auxiliary rewards requires the task to be decomposed into smaller subtasks, which may not be trivial. Further, they query the intention policies several times within a single episode. Unlike CAMS and SAC-X, which rely on selecting expert policies to perform sub-tasks, our approach trains an independent learner policy. It acquires expertise from sub-optimal experts using only a single oracle query per episode, thus having the potential to surpass these oracles through global exploration.

**Policy improvement with multiple experts.** Recent works attempt to learn from suboptimal black-box oracles while also utilizing rewards observed under the learner's policy. Active offline policy selection (A-OPS) (Konyushova et al., 2021) utilizes policy similarities to enhance value predictions. However, A-OPS lacks a learner policy to acquire expertise from these offline policies. ILEED (Beliaev et al., 2022) distinguishes between oracles based on their expertise at each state but is constrained to pure offline IL settings. InfoGAIL (Li et al., 2017) conditions the learned policy on latent factors that motivate demonstrations of different oracles. OIL (Li et al., 2018) tries to identify and follow the best oracle in a given situation. SFQL (Barreto et al., 2017) proposes *generalized policy improvement* with successor features. MAMBA (Cheng et al., 2020) utilizes an advantage function with geometric weighted generalization and achieves a larger policy improvement over SFQL, while addressing the above two important questions with theoretical support. MAPS (Liu et al., 2023) improves on the sample efficiency and performance of MAMBA by proposing active policy selection and state exploration. However, even when the quality of the oracle set is poor, these algorithms will still resort to imitation learning with the inferior oracles. In contrast, our algorithm performs self-improvement, employing imitation learning only on states for which an oracle outperforms the learner.

## 3 PRELIMINARIES

We consider a finite-horizon Markov decision process (MDP) $\mathcal{M}_0 = \langle \mathcal{S}, \mathcal{A}, \mathcal{P}, r, H \rangle$ with state space $\mathcal{S}$, action space $\mathcal{A}$, unknown stochastic transition dynamics $\mathcal{P} : \mathcal{S} \times \mathcal{A} \to \Delta(\mathcal{S})$, unknown reward function $r : \mathcal{S} \times \mathcal{A} \to [0, 1]$, and episode horizon $H$. We define total number of training steps (rounds) as $N$ and assume access to a (possibly empty) set of $K$ oracles, defined as $\Pi = \{\pi^k\}_{k=1}^K$, where $\pi_k : \mathcal{S} \to \Delta(\mathcal{A})$. The *generalized Q-function* with respect to a general function $f : \mathcal{S} \to \mathbb{R}$ is defined as:

$$Q^f(s, a) := r(s, a) + \mathbb{E}_{s' \sim \mathcal{P}|s,a}[f(s')].$$

When $f(s)$ is the value function of a particular policy $\pi$, the generalized Q-function can be used to recover the policy's Q-function $Q^\pi(s, a)$. We denote the *generalized advantage function* with respect to $f$ as

$$\boldsymbol{A}^f(s, a) = Q^f(s, a) - f(s) = r(s, a) + \mathbb{E}_{s' \sim \mathcal{P}|s,a}[f(s')] - f(s).$$

Given an initial state distribution $d_0 \in \Delta(\mathcal{S})$, let $d_t^\pi$ denote the distribution over states at time $t$ under policy $\pi$. The state visitation distribution under $\pi$ can be expressed as $d^\pi := \frac{1}{H} \sum_{t=0}^{H-1} d_t^\pi$. The value function of the policy $\pi$ under $d_0$ is denoted as:

$$V^\pi(d_0) = \mathbb{E}_{s_0 \sim d_0}[V^\pi(s)] = \mathbb{E}_{s_0 \sim d_0}\left[\mathbb{E}_{\tau_0 \sim \rho^\pi|s_0}\left[\sum_{t=0}^{H-1} r(s_t, a_t)\right]\right]$$

where $\rho^\pi(\tau_t \mid s_t)$ is the distribution over trajectories $\tau_t = \{s_t, a_t, \ldots, s_{H-1}, a_{H-1}\}$ under policy $\pi$. The goal is to find a policy $\pi = \arg\max_\pi J(\pi)$ maximizing the expected return

$$J(\pi) = \mathbb{E}_{s \sim d_0}[V^\pi(s)]. \tag{1}$$

## 4 POLICY IMPROVEMENT WITH PERFECT KNOWLEDGE OF ORACLE SET

We now present a reinforcement learning framework in the presence of an imitation learning oracle set, which is inspired from Cheng et al. (2020); Liu et al. (2023). In this section, we assume that we have perfect knowledge of the underlying MDP and each oracle's value function. We will relax these assumptions in the next section.

**Max-following.** Given a collection of $k$ imitation learning *oracles* $\Pi^o = \{\pi^k\}_{k \in [K]}$, the *max-following* policy is a greedy policy that selects the oracle with the highest expertise in any given state. The max-following policy is sensitive to the quality of the oracles. Specifically, if all oracles perform worse than the learner policy at a given state, the max-following policy will still naively imitate the best (but poor) oracle. Instead, it would be more prudent to follow the learner's guidance in these cases.

**Definition 4.1. (Extended Oracle Set).** Let $\Pi^o = \{\pi^k\}_{k \in [K]}$ be the given black-box oracle set, $\Pi^\mathcal{L} = \{\pi_n\}_{n \in [N]}$ be the learner's policy class, where $\pi_n$ denotes that the policy has been updated for $n$ rounds. We define the *extended oracle set* at the $n$-th round as

$$\Pi^\mathcal{E} = \Pi^o \cup \{\pi_n\} = \{\pi^1, \ldots, \pi^K, \pi_n\}. \tag{2}$$

**Remark 4.2.** The learner policy in the extended oracle set is updated at each round.

### 4.1 MAX$^+$ AGGREGATION

Based on the extended oracle set, we first introduce the advantage function $\boldsymbol{A}^+$ and the baseline value function $f^+$ as follows:

**Definition 4.3. ($A^+$ Advantage Function).** Given $k$ oracles $\pi^1, \ldots, \pi^k$ and the learner policy $\pi_n$, we define $\boldsymbol{A}^+$ advantage function as :

$$\boldsymbol{A}^+(s, a) := r(s, a) + \mathbb{E}_{s' \sim \mathcal{P}|\pi,s}[f^+(s')] - f^+(s), \tag{3}$$

where $f^+(s)$ is the baseline value function, defined as:

$$f^+(s) = \max_{k \in [|\Pi^\mathcal{E}|]} V^k(s), \text{ where } [V^k]_{k \in [|\Pi^\mathcal{E}|]} := \left[V^{\pi^1}, \ldots, V^{\pi^K}, V^{\pi_n}\right]. \tag{4}$$

$f^+(s)$ focuses exclusively on optimizing oracle selection for a single state, assuming that the selected policy will be followed for the remainder of the trajectory. To optimize the oracle selection for every encountered state, we introduce the max$^+$-following policy, which acts as a greedy policy, adhering to the optimal policy within the *extended* oracle set for any given state.

**Definition 4.4.** (**Max$^+$-Following Policy**). Given extended oracle set $\Pi^{\mathcal{E}}$, the *max$^+$-following* policy

$$\pi^{\circ}(a \mid s) := \pi^{k^{\star}}(a \mid s), \text{ where } k^{\star} := \arg\max_{k \in [|\Pi^{\mathcal{E}}|]} V^k(s), |\Pi^{\mathcal{E}}| = K+1, V^{K+1} = V^{\pi_n}. \quad (5)$$

**Proposition 4.5.** *Following $\pi^{\circ}$ is as good or better than imitating the single-best policy in $\Pi^{\mathcal{E}}$.*

With a slight abuse of notation, we use $\boldsymbol{A}^+(s, \pi^{\circ})$ to denote the generalized advantage function of the policy $\pi^{\circ}$ at $s$. As proved in the Appendix, the function $\boldsymbol{A}^+(s, \pi^{\circ}) \geq 0$ (Appendix C.1) and that the value function for the max$^+$-following policy satisfies $V^{\pi^{\circ}}(s) \geq f^+(s) = \max_{k \in [|\Pi^{\mathcal{E}}|]} V^k(s)$ (Appendix C.1). This indicates that following $\pi^{\circ}$ is at least as good as or better than imitating a single best policy in $\Pi^{\mathcal{E}}$. Thus, $\pi^{\circ}$ is a valid approach to robust policy learning in the multiple oracle setting. The max$^+$-following policy $\pi^{\circ}$ is better than the max-following policy when the learner's policy is better than any oracle for a given state. On the other hand, when the value of a specific oracle $V^k$ is always better than all other policies for all states, $\pi^{\circ}$ simply reduces to the corresponding oracle $\pi^k$. This is not ideal because the value $V^k(s)$ assumes to keep rolling out the same oracle $\pi^k$ from state $s$ until termination, without making improvement by looking one step ahead and searching for a better action. To address this, we propose the *max$^+$-aggregation* policy as follows.

**Definition 4.6.** (**Max$^+$-Aggregation Policy**[2]). For state $s$, the max$^+$-aggregation policy $\pi^{\circledcirc}$ performs one-step improvement and takes the action with largest advantage over $f^+$,

$$\pi^{\circledcirc}(a \mid s) = \delta_{a=a^{\star}}, \text{where } a^{\star} = \arg\max_{a \in \mathcal{A}} \boldsymbol{A}^+(s, a) \text{ and } \delta \text{ is the Dirac delta distribution.} \quad (6)$$

Although the max$^+$-following policy $\pi^{\circ}$ improves upon the max-following policy, it does not perform self-improvement. In contrast, the max$^+$-aggregation policy $\pi^{\circledcirc}$ looks one step ahead and makes the largest one-step advantage improvement with respect to $f^+$. Thus, in the degenerate case where $\pi^{\circ}$ is equivalent to the single best policy, $\pi^{\circledcirc}$ outperforms the best single policy in $\Pi^{\mathcal{E}}$ for all states. Since $\boldsymbol{A}^+(s, \pi^{\circledcirc}) \geq \boldsymbol{A}^+(s, \pi^{\circ}) \geq 0$ for any state $s$ by Corollary C.2 and Proposition 4.5, we conclude that the max$^+$-aggregation policy $\pi^{\circledcirc}$ is a suitable policy benchmark for the robust policy learning setting as well. We note that the baseline $f^+(s)$ corresponds to the value of choosing the single-best policy in $\Pi^{\mathcal{E}}$ at state $s$ and rolling it out throughout the rest of the episode. In contrast, $\pi^{\circ}$ and $\pi^{\circledcirc}$ optimize the oracle selection at every remaining step in the trajectory. This work is therefore built on $\pi^{\circledcirc}$.

**Remark 4.7.** (**Empty Oracle Set**) Up to this point, we have primarily assumed a non-empty oracle set $\Pi^o$ and an extended oracle set of size $|\Pi^{\mathcal{E}}| \geq 2$. Given an empty oracle set $\Pi^o$, $\Pi^{\mathcal{E}}$ will only contain the learner policy. In this case, $f^+ \equiv V^{\pi^{\circ}}$ and $\pi^{\circ}$ will not improve, while $\pi^{\circledcirc}$ reduces to pure reinforcement learning, performing self-improvement by using the advantage function $\boldsymbol{A}^+$.

## 5 ROBUST POLICY IMPROVEMENT WITH BLACK-BOX ORACLE SET

Improving a policy from the max$^+$ baseline $f^+$ (Eqn. 4) is the key to learning robustly via IL and RL. This requires knowledge of the MDP and the oracles' value functions, however, the oracles are presented to the learner as black-box policies with unknown value functions.

A critical challenge to use $f^+(s) = \max_{k \in [|\Pi^{\mathcal{E}}|]} V^k(s)$ as a baseline is that it changes as training goes, whereas MAMBA assumes a static baseline function. In the following analysis we resort to a slightly weaker baseline, $f_m^+ := \max_{k \in [|\Pi^o \cup \{\pi_m\}|]} V^k(s)$, where $m \ll N$ is an intermediate step in the learning process, and $N$ is the total number of rounds. Similarly, we define $\boldsymbol{A}_m^+(s, a) := r(s, a) + \mathbb{E}_{s' \sim \mathcal{P}|\pi,s}[f_m^+(s')] - f_m^+(s)$, as the corresponding advantage function, and $\pi_m^{\circledcirc}$ as the corresponding max$^+$-aggregation policy by setting $\boldsymbol{A}^+ = \boldsymbol{A}_m^+$ in Definition 4.6. In the following, we use the baseline value $f_m^+$, and reformulate the problem in an online learning setting (Ross et al., 2011; Ross and Bagnell, 2014; Sun et al., 2017; Cheng et al., 2020; Liu et al., 2023) for black-box oracles. Following MAMBA's analysis, we first assume that the oracle value functions are known but the MDP is unknown, followed by the case that the value functions are unknown.

---

[2]When we exclude the learner's policy from the extended oracle set, this reduces to the max-aggregation policy, which was used in MAMBA (Cheng et al., 2020).

**Unknown MDP with known value functions.** If the MDP is unknown, we can regard $d^{\pi_n}$ as an adversary in online learning and establish the online loss for round $n$ as

$$\ell_n(\pi) := -H\mathbb{E}_{s \sim d^{\pi_n}} \mathbb{E}_{a \sim \pi|s} [\boldsymbol{A}^+(s,a)]. \tag{7}$$

Lemma C.1 and Proposition 4.5 suggest that making $\ell_n(\pi)$ small ensures that $V^{\pi_n}(d_0)$ achieves better performance than $f_m^+(d_0)$ for $m < n$. Averaging over $N$ rounds of online learning, we obtain

$$\frac{1}{N} \sum_{n \in [N]} V^{\pi_n}(d_0) = f_m^+(d_0) + \Delta_N - \epsilon_N(\Pi^{\mathcal{L}}) - \text{Regret}_N^{\mathcal{L}}, \tag{8}$$

where $\text{Regret}_N^{\mathcal{L}} := \frac{1}{N}(\sum_{n=1}^N \ell_n(\pi_n) - \min_{\pi \in \Pi^{\mathcal{L}}} \sum_{n=1}^N \ell_n(\pi))$ depends the learning speed of an online algorithm, $\Delta_N := -\frac{1}{N}\sum_{n=1}^N \ell_n(\pi_m^{\circledcirc})$ is the loss of the baseline max$^+$-aggregation policy $\pi_m^{\circledcirc}$, and $\epsilon_N(\Pi^{\mathcal{L}}) := \min_{\pi \in \Pi^{\mathcal{L}}} \frac{1}{N}(\sum_{n=1}^N \ell_n(\pi) - \sum_{n=1}^N \ell_n(\pi_m^{\circledcirc}))$ expresses the quality of oracle class, where $\Pi^{\mathcal{L}}$ is specified in Definition 4.1. If $\pi_m^{\circledcirc} \in \Pi^{\mathcal{L}}$, we have $\epsilon_N(\Pi^{\mathcal{L}}) = 0$. Otherwise, $\epsilon_N(\Pi^{\mathcal{L}}) > 0$. By Proposition 4.5, $\boldsymbol{A}^+(s, \pi_m^{\circledcirc}) \geq 0$ and, in turn, $\Delta_N \geq 0$. If $\pi^{\circledcirc} \in \Pi^{\mathcal{L}}$, using a no-regret algorithm to address this online learning problem will produce a policy that achieves performance of at least $\mathbb{E}_{s \sim d_0}[f_m^+(s)] + \Delta_N + O(1)$ after $N$ iterations.

**Unknown MDP with unknown value function.** In practice, the value functions of the oracle set are unavailable. $f^+$ and $\boldsymbol{A}^+$ need to be approximated by $\hat{f}^+$ and $\hat{\boldsymbol{A}}^+$. We compute the sample estimate of the gradient as follows:

$$\nabla \hat{\ell}_n(\pi_n) = -H\mathbb{E}_{s \sim d^{\pi_n}} \mathbb{E}_{a \sim \pi_n|s} \left[ \nabla \log \pi_n(a \mid s) \hat{\boldsymbol{A}}^+(s,a). \right] \tag{9}$$

The approximation of the value function and gradient introduces bias and variance terms in the online learning regret bound $\text{Regret}_N^{\mathcal{L}}$. We propose a general theorem to lower bound the performance:

**Proposition 5.1** (Adapted from Cheng et al. (2020)). *Define $\Delta_N$, $\epsilon_N(\Pi^{\mathcal{L}})$, $f_m^+$, and $\text{Regret}_N^{\mathcal{L}}$ as above, where $f_m^+ := \max_{k \in [|\Pi^o \cup \{\pi_m\}|]} V^k(s)$ for $m \leq N$, and $\text{Regret}_N^{\mathcal{L}}$ corresponds to the regret of a first-order online learning algorithm based on Eqn. 9. It holds that*

$$\mathbb{E}\left[\max_{n \in [N]} V^{\pi_n}(d_0)\right] \geq \mathbb{E}_{s \sim d_0}[f_m^+(s)] + \mathbb{E}\left[\Delta_N - \epsilon_N(\Pi^{\mathcal{L}}) - \text{Regret}_N^{\mathcal{L}}\right],$$

*where the expectation is over the randomness in feedback and the online algorithm.*

**Remark 5.2.** W.l.o.g. we assume $f_0^+(s) := \arg\max_{k \in [K]} V^k(s)$, which corresponds to the baseline function considered in MAMBA. Note that $f_m^+(s)$ admits a weaker baseline value than $f_n^+(s)$ for $m < n$, but *no weaker than* the max value of any oracle, $f_0^+(s)$. Therefore, as the learner improves $f_m^+(s)$, max$^+$-aggregation will have an improved lower bound over Cheng et al. (2020). Consider a scenario where $m = o(N)$. In round $m$, we instantiate $f_m^+$ and perform 1-step advantage improvement over $f_m^+$. Since $f_m^+(s) > f_0^+(s)$ when $V^{\pi_m(s)} > f_0^+(s), s \sim d^{\pi_n}$, we can view max$^+$-aggregation as adding improved learner policies into $\Pi^o$ at the end of each round and perform 1-step improvement over $f^+$ on the *expending* oracle set. As $\mathbb{E}_{s \sim d_0}[f_m^+(s)]$ improves, it will lead to the improvement over the original bound in Proposition 5.1.

## 6 ROBUST POLICY IMPROVEMENT VIA ACTIVELY BLENDING RL AND IL

In this section, we present RPI, an algorithm for robust policy improvement that builds upon the max$^+$-aggregation policy. RPI consists of two main components: Robust Active Policy Selection (RAPS) and Robust Policy Gradient (RPG) that enable the algorithm to combine the advantages of reinforcement and imitation learning.

### 6.1 ROBUST ACTIVE POLICY SELECTION

To improve the sample efficiency in learning from multiple oracles and lower the bias in $\text{Regret}_N^{\mathcal{L}}$ in Proposition 5.1 caused by the approximator of the max$^+$ baseline function $\hat{f}^+$, we propose a *robust active policy selection* strategy. We employ an ensemble of prediction models to estimate the value

---

**Algorithm 1** Robust Policy Improvement (RPI)

---

**Input:** Learner policy $\pi_1$, oracle set $\Pi = \left\{\pi^k\right\}_{k \in [K]}$, function approximators $\{\hat{V}^k\}_{k \in [K]}, \hat{V}_n$.

**Output:** The best policy among $\{\pi_1, ..., \pi_N\}$.

1: **for** $n = 1, \ldots, N-1$ **do**
2:    Construct an extended oracle set $\Pi^{\mathcal{E}} = \left[\pi^1, \ldots, \pi^k, \pi_n\right]_{k \in [|\Pi|]}$.
3:    Sample $t_e \in [H-1]$ uniformly random.
4:    Roll-in $\pi_n$ up to $t_e$, select $k_\star$ (Eqn. 10), and roll out $\pi^{k_\star}$ to collect the remaining data $\mathcal{D}^k$.
5:    Update $\hat{V}^{k_\star}$ using $\mathcal{D}^k$.
6:    Roll-in $\pi_n$ for full $H$-horizon to collect data $\mathcal{D}'_n$.
7:    Update $\hat{V}_n$ using $\mathcal{D}'_n$.
8:    Compute advantage $\hat{A}^{\text{GAE+}}$ (Eqn. 11) and gradient estimate $\hat{g}_n$ (Eqn. 14) using $\mathcal{D}'_n$.
9:    Update $\pi_n$ to $\pi_{n+1}$ by giving $\hat{g}_n$ to a first-order online learning algorithm.

---

function for a policy (Liu et al., 2023), where we estimate both the mean $\hat{V}^k_\mu(s)$ and the uncertainty $\sigma_k(s)$ for a particular state $s$. We generate a few independent value prediction networks that are initialized randomly, and then train them using random samples from the trajectory buffer of the corresponding oracle $\pi^k$.

In the single oracle case, the motivation of rolling in a learner policy and rolling out an oracle policy (referred to RIRO) in prior work (e.g., DAgger, AggrevateD) is to address the distribution shift. In our work, in addition to addressing distribution shift, we aim to improve the value function estimator $\hat{V}$ of the most promising oracle on the switch state $s$ to reduce the bias term of $\hat{f}^+$. Moreover, we seek to reduce the roll-out cost associated with querying oracles, particularly when the learner exhibits a higher expected value for the switching state. In such cases, we roll-out the learner to collect additional data to enhance its policy. We achieve this goal by comparing the UCB of oracle policies' value function and LCB of learner policy to improve the estimation of $\hat{f}^+$. We design the strategy as follows:

Let $\overline{\hat{V}^k}(s) = \hat{V}^k_\mu(s) + \sigma_k(s), \underline{\hat{V}^k}(s) = \hat{V}^k_\mu(s) - \sigma_k(s)$ be the UCB and LCB of policy $k$'s value function for state $s$, respectively. We obtain the best oracle $\pi^{k_\star}$ for state $s$ as follows:

$$k_\star = \underset{k \in [|\Pi^{\mathcal{E}}|]}{\arg\max} \left\{ \overline{\hat{V}^1}(s), \overline{\hat{V}^2}(s), ..., \overline{\hat{V}^K}(s), \underline{\hat{V}^{K+1}}(s) \right\}, \quad (10)$$

where $\underline{\hat{V}^{K+1}}$ is the confidence-aware value function approximator for the learner's policy, while $[\hat{V}^k]_{k \in [K]}$ represents the value function approximators associated with oracle policies.

**Remark 6.1.** The insight behind using a confidence-aware policy selection strategy in RAPS is to improve the estimate of the value function of the most promising oracle at a given state. This necessitates accounting for estimation uncertainties, which leads to the adoption of a UCB-based approach to identify the optimal oracle. Using LCB for the learner encourages oracle-guided exploration unless we are certain that the learner surpasses all oracles for the given state. We empirically evaluate this in Section 7.2.

**Remark 6.2.** MAPS (Liu et al., 2023) introduced an active policy selection strategy by selecting the best oracle to roll out and improve the value function approximation on state $s_{t_e}$ according to $f^+_0$. In this work, we empirically improve such strategy by utilizing the learner policy in $\Pi^{\mathcal{E}}$.

## 6.2 ROBUST POLICY GRADIENT

We now propose robust policy gradient based on a novel advantage function, denoted by $A^{\text{GAE+}}$ and a novel max$^+$ actor-critic framework.

$A^{\text{GAE+}}$ **advantage function.** The policy gradient methods maximize the expected total reward by repeatedly estimating the gradient $g := \nabla_\theta \mathbb{E}[\sum_{t=0}^{H-1} r_t]$. The policy gradient has the form $g = \mathbb{E}_t[\nabla_\theta \log \pi_\theta(a_t|s_t)\hat{A}_t]$ (Sutton et al., 1999; Greensmith et al., 2004; Schulman et al., 2015; 2017), where $\pi_\theta$ is a stochastic learner policy and $\hat{A}_t$ is an estimator of the advantage function at timestep $t$ and $\mathbb{E}[\cdot]$ indicates the empirical average over a finite batch of samples, for an algorithm that

Figure 1: We consider 8 tasks from DeepMind Control Suite and Meta-World. Extended results on different variants of these tasks are provided in Appendices E.2 & E.3.

alternates between sampling and optimization. $\boldsymbol{A}_t$ measures whether the action is better or worse than the current policy. Hence, the gradient term $\nabla_\theta \log \pi_\theta (a_t|s_t) \hat{\boldsymbol{A}}_t$ points in the direction of increased $\pi_\theta (a_t|s_t)$ if and only if $\hat{\boldsymbol{A}}_t = \hat{\boldsymbol{A}} (s_t, a_t) > 0$. For $\hat{\boldsymbol{A}}$, we propose a novel advantage function $\boldsymbol{A}^{\text{GAE+}}$ based on general advantage estimation (Schulman et al., 2015), the max$^+$ baseline $f^+$ and the $\boldsymbol{A}^+$ advantage function (3).

$$\hat{\boldsymbol{A}}_t^{\text{GAE}(\gamma,\lambda)+} = \hat{\delta}_t + (\gamma\lambda)\hat{\delta}_{t+1} + ... + (\lambda\gamma)^{T-t+1}\hat{\delta}_{T-1}, \text{ where } \hat{\delta}_t = r_t + \gamma\hat{f}^+ (s_{t+1}) - \hat{f}^+ (s_t), \quad (11)$$

where $T \ll H$, and $\gamma$ and $\lambda$ are the predefined parameters that control the bias-variance tradeoff. In this work, we use $\lambda = 0.9$ and $\gamma = 1$, and thus simplify $\hat{A}_t^{\text{GAE}(\gamma,\lambda)+}$ as $\hat{A}_t^{\text{GAE+}}$.

We propose a variant of the max$^+$ baseline $f^+$ that includes a confidence threshold $\Gamma_s$ for an oracle's value estimate:

$$\hat{f}^+ (s) = \begin{cases} \hat{V}_\mu^{\pi_n} (s), & \text{if } \sigma_k (s) > \Gamma_s, \text{where } k = \arg\max_{k \in [|\Pi^\varepsilon|]} \hat{V}_\mu^k (s). \\ \max_{k \in [|\Pi^\varepsilon|]} \hat{V}_\mu^k (s), & \text{otherwise.} \end{cases} \quad (12)$$

**Remark 6.3.** We use a threshold to control the reliability of taking the advice of an oracle, where a lower value indicates greater confidence. In our experiments, we use $\Gamma_s = 0.5$, which we have found to exhibit robust behavior (Appendix E.5).

Finally, we have the $n$-th round online loss as

$$\hat{\ell}_n (\pi_n) := -H\mathbb{E}_{s \sim d^{\pi_n}} \mathbb{E}_{a \sim \pi|s} \left[\hat{\boldsymbol{A}}^{\text{GAE+}} (s, a)\right]|_{\pi=\pi_n}, \quad (13)$$

and gradient estimator as

$$\hat{g}_n = \nabla\hat{\ell}_n (\pi_n) = -H\mathbb{E}_{s \sim d^{\pi_n}} \mathbb{E}_{a \sim \pi|s} \left[\nabla\log\pi (a \mid s) \hat{\boldsymbol{A}}_t^{\text{GAE+}} (s, a)\right]|_{\pi=\pi_n}. \quad (14)$$

**Max$^+$ actor-critic.** We note that the RPG component (Algorithm 1, lines 8–9) can be viewed as a variant of the *actor-critic* framework, with the actor sampling trajectories that are then evaluated by the max$^+$ critic based on the $\boldsymbol{A}^{\text{GAE+}}$ advantage function (11). The policy gradient in Eqn. 14 enables the learner policy $\pi_n$ to learn from high-performing oracles and to improve its own value function $\hat{V}^k$ for the states in which the oracles perform poorly.

**Remark 6.4.** When $\gamma = 1$, Eqn. 11 disregards the accuracy of $\hat{f}^+$, but it has high variance due to the sum of the reward terms. When $\gamma = 0$, it introduces bias, but has much lower variance. Moreover, when $\lambda = 0$ and $\gamma = 1$, the loss (13) of RPI reduced to the loss (7) under max$^+$-aggregation (6), and the performance bound for the max$^+$-aggregation policy and RPI will be equal. Thus, performing no-regret online learning with regards to Eqn. 13 has the guarantee in Proposition 5.1 and Remark 5.2. However, when $\lambda > 0$, RPI will optimize the multi-steps advantage over $f^+$ in Eqn. 13, while the max$^+$-aggregation policy $\pi^\odot$ only optimizes the one-step advantage over $f^+$. Thus, RPI will have a smaller $\epsilon_N (\Pi^{\mathcal{L}})$ term than max$^+$-aggregation, which improves the performance lower bound in Proposition 5.1.

**Imitation, Blending and Reinforcement.** Instances of $\hat{f}^+$ in Eqn. 11 may involve a combination of oracles' and learner's value functions. In a case that this does not involve the learner's value function—this is likely in the early stage of training since the learner's performance is poor—RPI performs imitation learning on the oracle policies. Once the learner policy improves and $\hat{f}^+$ becomes identical to the learner's value function, RPI becomes equivalent to the vanilla actor-critic that performs self-improvement. When it is a combination of the two, RPI learns from a blending of the learner and oracles.

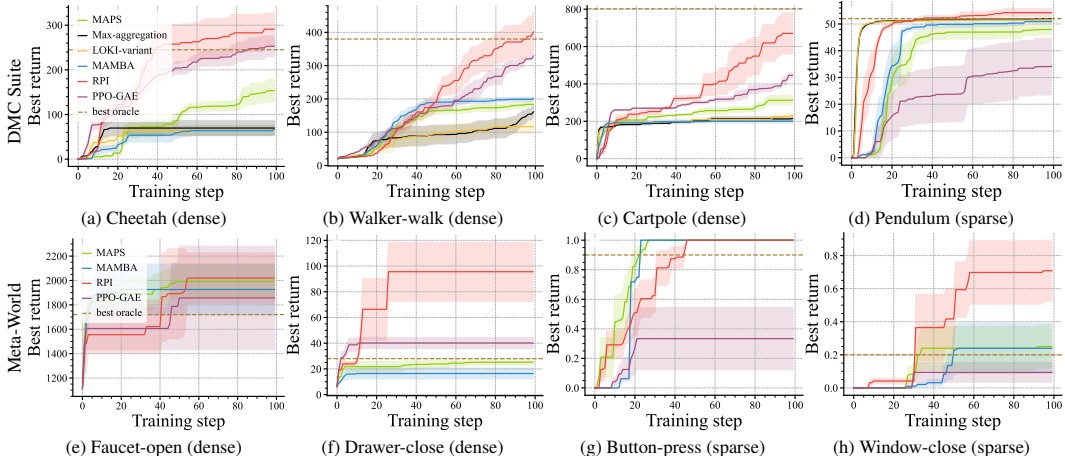

Figure 2: **Main results.** A comparison between RPI with five baselines and the best oracle (horizontal line) on Cheetah, Cartpole, Pendulum, and Walker-walk from DMC; and Window-close, Button-press, Faucet-open, and Drawer-close from Meta-World in terms of best-return with three diversified oracles. The shaded area represents the standard error over five trials. RPI scores the best in all benchmarks.

## 7 EXPERIMENTS

**Environments.** We evaluate our method on eight continuous state and action space domains: Cheetah-run, CartPole-swingup, Pendulum-swingup, and Walker-walk from the DeepMind Control Suite (Tassa et al., 2018); and Window-close, Faucet-open, Drawer-close and Button-press from Meta-World (Yu et al., 2020). In addition, we conduct experiments on a modified sparse reward Meta-World environment, which is considered to be a more challenging task. We set ensemble size of value functions to five. Appendix D provides further details.

**Oracles.** We implement our oracles as policies trained using PPO (Schulman et al., 2017) with generalized advantage estimate (GAE) (Schulman et al., 2015) and SAC (Haarnoja et al., 2018). We save the policy weights at different points during training to achieve oracles that perform differently in different states. Each environment is provided with three diversified oracles.

**Baselines.** We compare RPI with five baselines: (1) PPO with GAE as a pure RL baseline; (2) Max-Aggregation (Cheng et al., 2020) as a pure IL baseline (a multiple-oracle variant of AggreVaTe(D)); (3) a variant of LOKI adapted to the multiple-oracle setting that initially performs pure IL and then pure RL; (4) MAMBA; (5) MAPS (the current state-of-the-art method)[3]; and also the best oracle in the oracle set as a reference. We matched the number of environment interactions across algorithms[4]. Appendix D provides further details.

### 7.1 MAIN RESULTS

Figure 2 visualizes the performance of RPI and the baselines. The results show that RPI surpasses the baselines on all domains, despite variations in the black-box oracle set. Notably, the RL-based PPO-GAE baseline outperforms the IL methods in the later stages of training in most of the dense reward environments, while IL-based approaches perform better in the sparse reward domains . Pendulum-swingup (Fig. 2(d)) and window-close (Fig. 2(h)) are particularly difficult domains that involve non-trivial dynamics and sparse reward (i.e., the agent receives a reward of 1 only when the pole is near vertical, the window is closed exactly). Due to the sparse reward, the IL-based approaches are significantly more sample efficient than the RL-based approach, but their performance plateaus quickly. RPI initially bootstraps from the oracles, and due to their suboptimality, it switches to self-improvement (i.e., learning from its own value function), resulting in better performance than both IL and RL methods. These results demonstrate the robustness of RPI as it actively combines the advantages of IL and RL to adapt to various environment.

---

[3]Our experimental setup including the oracle set differs from that of MAPS. In this work, the learner for all baselines has access to approximately the same number of transitions and the learner does not have access to the oracle's trajectory. We reproduce the baseline performance for the MAPS' setting in Appendix E.1.

[4]PPO has a slightly smaller number of interactions due to the lack of oracles' value function pre-training.

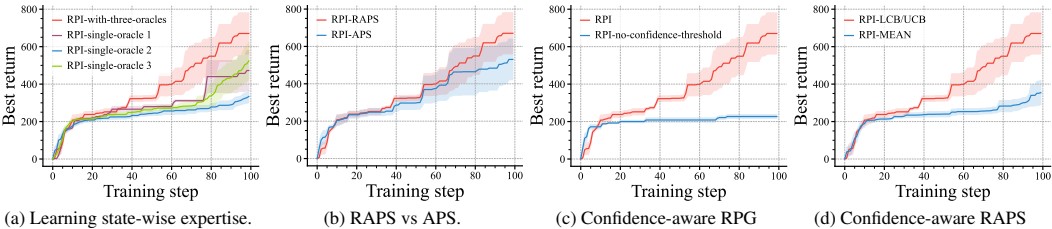

Figure 3: Results of ablation studies on the Cartpole environment.

## 7.2 ABLATION STUDIES

**Learning state-wise oracle expertise.** In Fig. 3(a), we examine the ability of RPI to aggregate the expertise of multiple oracles on Cartpole. We created three diversified oracles and find that RPI achieves a return of $645$ when it is able to query all three oracles, while the best return falls below $600$ when given access to only a single oracle. This result demonstrates RPI's utilization of the oracles' state-wise expertise, as it achieves better performance when given access to more oracles.

**Ablation on robust active policy selection.** In order to understand the effectiveness of RPI's robust active policy selection strategy (RAPS), we compare it to active policy selection (APS) (Liu et al., 2023) (without the learner in RIRO (Algorithm 1, line 4)) on Cartpole. Fig. 3(b) shows that RAPS has the advantage of selecting the learner policy to roll out in states for which it outperforms the oracles, resulting in self-improvement. This leads to RAPS outperforming the APS-based approach.

**Confidence-awareness in RPI.** *(1) RPG*: We first perform an ablation on Cartpole to investigate the benefits of using a confidence threshold on an oracle's value estimate for RPG (Eqn. 12). We see in Fig. 3(c) that the confidence threshold enables RPG to benefit from both state-wise imitation learning from oracles with high confidence and the execution of reinforcement learning when oracles exhibit high uncertainty. Without the threshold, RPG is more vulnerable to the quality of oracle set. *(2) RAPS*: We then consider the benefits of reasoning over uncertainty to the policy selection strategy, comparing uncertainty-aware RPI-LCB/UCB (Eqn. 10) to RPI-MEAN, which does not consider uncertainty. Fig. 3(d) demonstrates the benefits of using LCB/UCB for policy selection. Addition results in Appendix E.6 reveal that RPI-LCB/UCB outperforms RPI-MEAN across all benchmarks by an *overall* margin of 40%, supporting the advantage of incorporating confidence to policy selection.

**Visualizing active IL and RL.** Figure 4 visualizes the active state-wise imitation and reinforcement process employed by RPI in the gradient estimator on Pendulum. The figure includes three oracle policies (in blue, orange, and green) and the learner's policy (in red). Each oracle exhibits different expertise at different stages. In the beginning, RPI only imitates the oracles, which initially have greater state-wise expertise than the learner. As the learner improves, the frequency with which RPI samples the leaner policy increases, corresponding to self-improvement. As training continues, the expertise of the learner increasingly exceeds that of the oracles, resulting in RPI choosing self-improvement more often than imitating the oracles.

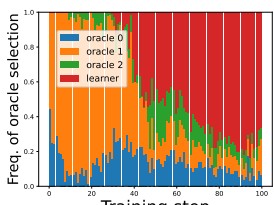

Figure 4: IL and RL.

## 8 CONCLUSION

We present max$^+$, a robust framework for IL and RL in the presence of a set of black-box oracles. Within this framework, we introduce RPI, a policy gradient algorithm comprised of two novel components: a robust active policy selection strategy (RAPS) that enhances sample efficiency and a robust policy gradient (RPG) for policy improvement. We provide a rigorous theoretical analysis of RPI, demonstrating its superior performance compared to the current state-of-the-art. Moreover, empirical evaluations on a diverse set of tasks demonstrate that RPI consistently outperforms all IL and RL baselines, even in scenarios with limited oracle information (favoring RL) or sparse rewards (favoring IL). RPI effectively adapts to the nature of the domain and the quality of the oracles by actively interleaving IL and RL. Our work introduces new avenues for robust imitation and reinforcement learning and encourages future research on addressing more challenging tasks in robust settings, such as handling missing state or oracle information.

ACKNOWLEDGEMENTS

We thank Ching-An Cheng for constructive suggestions. This work is supported in part by the RadBio-AI project (DE-AC02-06CH11357), U.S. Department of Energy Office of Science, Office of Biological and Environment Research, the Improve project under contract (75N91019F00134, 75N91019D00024, 89233218CNA000001, DE-AC02-06-CH11357, DE-AC52-07NA27344, DE-AC05-00OR22725), the Exascale Computing Project (17-SC-20-SC), a collaborative effort of the U.S. Department of Energy Office of Science and the National Nuclear Security Administration, the AI-Assisted Hybrid Renewable Energy, Nutrient, and Water Recovery project (DOE DE-EE0009505), and NSF HDR TRIPODS (2216899).

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

# A    SELECTIVE COMPARISON AGAINST RELATED WORKS

Table 2: A qualitative comparison of related algorithms. The publication years are included in parentheses for reference. Algorithms designed to fit a particular criterion are marked by "✓"; criteria that are not explicitly considered in the algorithm design are marked by "×".

| Algorithm | Criterion | Online | Stateful | Active | Interactive | Multiple oracles | Sample efficiency (in multiple oracles) | Robust |
|---|---|---|---|---|---|---|---|---|
| Behavioral Cloning (Pomerleau, 1988) | IL | × | ✓ | × | × | × | – | × |
| REINFORCE (Williams, 1992) (Sutton et al., 1999) | RL | ✓ | ✓ | × | × | × | × | – |
| SMILe (Ross and Bagnell, 2010) | IL | × | ✓ | × | × | × | – | × |
| DAgger (Ross et al., 2011) | IL | ✓ | ✓ | × | ✓ | × | – | × |
| PPO with GAE (Schulman et al., 2017) (Schulman et al., 2015) | RL | ✓ | ✓ | × | × | × | × | – |
| AggreVateD (Sun et al., 2017) | IL | ✓ | ✓ | × | ✓ | × | – | × |
| DQfD (Hester et al., 2018) | Offline → online RL | ✓ | ✓ | – | ✓ | × | – | – |
| THOR (Sun et al., 2018) | IL+RL | ✓ | ✓ | × | ✓ | × | – | × |
| LOKI (Cheng et al., 2018) | IL+RL | ✓ | ✓ | × | ✓ | × | – | × |
| SAC-X (Riedmiller et al., 2018) | RL | ✓ | ✓ | × | ✓ | ✓ | × | × |
| LEAQI (Brantley et al., 2020) | IL | ✓ | ✓ | ✓ | ✓ | × | – | × |
| MAMBA (Cheng et al., 2020) | IL+RL | ✓ | ✓ | × | ✓ | ✓ | × | × |
| A-OPS (Konyushova et al., 2021) | Policy Sel. | × | × | ✓ | × | ✓ | ✓ | – |
| ILEED (Beliaev et al., 2022) | IL | × | ✓ | × | × | ✓ | × | × |
| IQL (Kostrikov et al., 2021) | Offline → online RL | ✓ | ✓ | – | ✓ | × | – | – |
| CAMS (Liu et al., 2022a) | Model Sel. | ✓ | × | ✓ | × | ✓ | ✓ | ✓ |
| MoDem (Hansen et al., 2022) | Offline → online RL | ✓ | ✓ | – | ✓ | × | – | – |
| Hybrid RL (Song et al., 2022) | Online RL with offline data | ✓ | ✓ | – | ✓ | × | – | – |
| PEX (Zhang et al., 2023) | Offline → online RL | ✓ | ✓ | – | ✓ | × | – | – |
| TGRL (Shenfeld et al., 2023) | IL+RL | ✓ | ✓ | × | ✓ | × | – | × |
| LfGP (Ablett et al., 2023) | IL | ✓ | ✓ | × | ✓ | ✓ | × | × |
| MAPS (Liu et al., 2023) | IL+RL | ✓ | ✓ | ✓ | ✓ | ✓ | ✓ | × |
| **RPI (Ours)** | IL+RL | ✓ | ✓ | ✓ | ✓ | ✓ | ✓ | ✓ |

## A.1    ADDITIONAL NOTES ON RELATED WORK

MAMBA addressed the challenge of learning from multiple sub-optimal oracles and tackled two fundamental questions: what constitutes a reasonable benchmark for policy improvement, and how to systematically combine sub-optimal oracles into a stronger baseline. MAMBA proposed a max-aggregated baseline and suggested policy improvement from it as a natural strategy for combining these oracles to form a better policy. In addition, they introduced a novel Max-aggregation of Multiple Baseline approach and provided a theoretical performance guarantee for it. However, one limitation of MAMBA is its high sample complexity. It requires prolonged rounds to identify the optimal oracle for a given state due to its strategy of uniformly sampling an oracle, resulting in a larger

accumulation of regret. MAPS aims to enhance the sample efficiency of MAMBA by introducing Max-aggregation Active Policy Selection with theoretical support, and it empirically outperforms MAMBA. Nevertheless, both MAMBA and MAPS share a common limitation in non-robustness. They are susceptible to the quality of the oracle set, and both algorithms may fail in cases where the oracle set is of poor quality. Our work addresses this robustness challenge by proposing a novel $\max^+$ framework. Inspired by the max-aggregation policy from MAMBA, we introduced a $\max^+$-aggregation policy based on a novel extended oracle set. This enables the policy to undergo self-improvement even when the oracle set is poor. Additionally, we proposed a novel algorithm, RPI, with Robust Active Policy Selection to improve active policy selection from MAPS. Theoretical analyses were provided for both the $\max^+$ framework and the RPI algorithm.

## B  ADDITIONAL BACKGROUND

### B.1  ADDITIONAL ALGORITHMS FOR LEARNING FROM MULTIPLE ORACLES

In this section, we introduce a few baselines that learn from a set of black-box oracles $\Pi = \left\{\pi^k\right\}_{k\in[K]}$.

**Single-best expert $\pi^\star$:** The first baseline that we consider imitates a single oracle that achieves the best performance in hindsight among the oracle set, i.e., $\pi^\star := \arg\max_{\pi\in\Pi} \mathbb{E}_{s_0\sim d_0}\left[V^\pi(s_0)\right]$. After figuring out the single-best expert, this strategy simply keeps rolling out the expert. In practice, this is often inadequate as it neglects the potential benefits of suboptimal oracles at the state level.

**Max-following $\pi^\bullet$:** Given a collection of $k$ imitation learning *oracles* $\Pi^\text{o} = \left\{\pi^k\right\}_{k\in[K]}$, the *max-following* policy (Cheng et al., 2020; Liu et al., 2023) is a greedy policy that selects the oracle with the highest expertise in any given state:

$$\pi^\bullet\left(a \mid s\right) := \pi^{k^\star}\left(a \mid s\right), \quad k^\star := \arg\max_{k\in[K]} V^k\left(s\right)$$

where $V^k(s) = V^{\pi^k}(s)$ is the value function for oracle $k \in [K]$.

**Max-aggregation $\pi^\text{max}$:** Max-aggregation (Cheng et al., 2020) performs one-step improvement based on the *max-following* policy $\pi^\bullet$. Denote a natural value baseline $f^\text{max}\left(s_t\right)$ for IL with multiple oracles as

$$f^\text{max}\left(s_t\right) := \max_{k\in[K]} V^k\left(s\right). \tag{15}$$

We then denote the *max-aggregation* policy as

$$\pi^\text{max}\left(a \mid s\right) := \delta_{a=a^\star}, \text{where } a^\star = \arg\max_{a\in\mathcal{A}} \boldsymbol{A}^{f^\text{max}}\left(s,a\right),$$

$$\boldsymbol{A}^{f^\text{max}}\left(s,a\right) = r\left(s,a\right) + \mathbb{E}_{s'\sim\mathcal{P}|s,a}[f^\text{max}\left(s'\right)] - f^\text{max}\left(s\right), \text{ and } \delta \text{ is the Dirac delta distribution.} \tag{16}$$

The max-aggregation policy is a function of $f^\text{max}$ and, in turn, requires knowledge of the MDP and each oracle's value function (Eqn. 15). However, in the episodic interactive IL setting, oracles are provided as black boxes and their value functions are unknown. MAMBA (Cheng et al., 2020) and MAPS (Liu et al., 2023) deal with this by reducing IL to an online learning problem and adapt the online loss defined at round $n$ as:

$$\ell_n\left(\pi;\lambda\right) := -\left(1-\lambda\right)H\mathbb{E}_{s\sim d^{\pi^n}}\left[\boldsymbol{A}_\lambda^{f^\text{max},\pi}\left(s,\pi\right)\right] - \lambda\mathbb{E}_{s\sim d_0}\left[\boldsymbol{A}_\lambda^{f^\text{max},\pi}\left(s,\pi\right)\right]. \tag{17}$$

Here, $\boldsymbol{A}_\lambda^{f^\text{max},\pi}(s,a)$ is a $\lambda$-weighted advantage defined as:

$$\boldsymbol{A}_\lambda^{f^\text{max},\pi}\left(s,a\right) := \left(1-\lambda\right)\sum_{i=0}^\infty \lambda^i \boldsymbol{A}_{(i)}^{f^\text{max},\pi}\left(s,a\right), \tag{18}$$

which integrates various $i$-step advantages:

$$\boldsymbol{A}_{(i)}^{f^\text{max},\pi}\left(s_t,a_t\right) := \mathbb{E}_{\tau_t\sim\rho^\pi(\cdot|s_t)}[r(s_t,a_t) + \cdots + r\left(s_{t+i},a_{t+i}\right) + f^\text{max}\left(s_{t+i+1}\right))] - f^\text{max}\left(s_t\right).$$

### B.2 LIMITATIONS OF THE PRIOR ART

MAPS (Liu et al., 2023) and MAMBA (Cheng et al., 2020) suffer from two limitations related to their non-robustness to the choice of the oracle set. First, their online loss function (17) relies on a predetermined $\lambda$ value that combines imitation learning and reinforcement learning, making their performance sensitive to the quality of the oracle set. Second, their gradient estimator utilizes the max-aggregation policy $\pi^{\max}$ and the value baseline function $f^{\max}$, both of which are dependent on the given black-box oracle set. If the oracle set includes only adversarial oracles, these methods will still try to perform imitation learning, thereby impeding policy enhancement.

### B.3 VALUE FUNCTION APPROXIMATOR FOR DISCRETE ENVIRONMENT

In the interactive episodic MDP, we roll out a selected oracle $k$, resulting in $N_k(s_t)$ trajectories $\tau_{1,k}, \tau_{2,k}, \ldots, \tau_{N_k,k}$ starting from state $s_t$ for round $N$. We determine an estimate for the return in state $s_t$ by averaging the returns obtained across the trajectories:

$$\hat{V}^{\pi_k}(s_t) = \frac{1}{N_k(s_t)} \sum_{i=1}^{N_k(s_t)} \sum_{j}^{H} \lambda^j r(s_j, a_j). \tag{19}$$

### B.4 ACTIVE POLICY SELECTION

To address the sample efficiency challenge in learning from multiple experts, we reference active policy selection technique in MAPS work to select the best oracle $k^\star$ for state $s_t$ as follows:

$$k_\star = \arg\max_{k \in [K]} \begin{cases} \hat{V}^k(s_t) + \sqrt{\frac{2H^2 \log \frac{2}{\delta}}{N_k(s_t)}} & \mathcal{S} \text{ discrete} \\ \hat{V}^k_\mu(s_t) + \sigma_k(s_t) & \mathcal{S} \text{ continuous} \end{cases} \tag{20}$$

## C PROOFS

In the following, we provide proofs for the theoretical claims in the main paper.

**Lemma C.1.** *(Kakade and Langford, 2002; Ng et al., 1999) Let $f : \mathcal{S} \to \mathbb{R}$ such that $f(s_H) = 0$. For any MDP and policy $\pi$,*

$$V^\pi(d_0) - f(d_0) = H\mathbb{E}_{s \sim d^\pi}\left[\mathbf{A}^f(s, \pi)\right] \tag{21}$$

From Lemma C.1, we get the following corollary:

**Corollary C.2.** *(Cheng et al., 2020) If $f$ is improvable with respect to $\pi$, then $V^\pi(s) \geq f(s)$, $\forall s \in \mathcal{S}$.*

Corollary C.2 indicates that a policy $\pi$ outperforms all policies in $\Pi^\mathcal{E}$, if, for every state, there is a baseline value function $f$ superior to that of all policies $(f(s) \geq V^k(s), \forall k \in [|\Pi^\mathcal{E}|], s \in \mathcal{S})$, while $f$ can be improved by $\pi$ (i.e., $\mathbf{A}^f(s, \pi) \geq 0$).

### C.1 PROOF OF PROPOSITION 4.5

*Proof.* Without loss of generality, let us assume the optimal oracle is oracle 1 (the first oracle) in oracle set $\Pi$,

$$\mathbf{A}^+(s, \pi^\circ) = r(s, \pi^\circ) + \mathbb{E}_{a \sim \pi^\circ|s}\mathbb{E}_{s' \sim \mathcal{P}|s,a}\left[f^+(s')\right] - f^+(s) \tag{22a}$$

$$\geq r(s, \pi^\circ) + \mathbb{E}_{a \sim \pi^\circ|s}\mathbb{E}_{s' \sim \mathcal{P}|s,a}\left[V^1(s')\right] - V^1(s) \tag{22b}$$

$$\geq r(s, \pi^\bullet) + \mathbb{E}_{a \sim \pi^\circ|s}\mathbb{E}_{s' \sim \mathcal{P}|s,a}\left[V^1(s')\right] - V^1(s) \tag{22c}$$

$$= \mathbf{A}^{V^1}(s, \pi^1) \geq 0, \tag{22d}$$

where the last step follows since $\pi^\circ(a|s) \geq \pi^\bullet(a|s) = \pi^1(a|s)$. Since we have $\boldsymbol{A}^+(s, \pi^\circ) \geq 0$, together with Lemma C.1, we have

$$V^{\pi^\circ}(s) \geq f^+(s) = \max_{k \in [|\Pi^{\mathcal{E}}|]} V^k(s). \tag{23}$$

$V^{\pi^\circ}(s) \geq f^+(s)$ indicates that following $\pi^\circ$ is equally good or superior to imitating a single best policy in $\Pi^{\mathcal{E}}$. □

## C.2 Proof of Proposition 5.1

We denote $f_0^+(s) := \arg\max_{k \in [K]} V^k(s)$. According to Theorem 1 of Cheng et al. (2020), we obtain

$$\mathbb{E}\left[\max_{n \in [N]} V^{\pi_n}(d_0)\right] \geq \mathbb{E}_{s \sim d_0}[f_0^+(s)] + \mathbb{E}\left[\Delta_N - \epsilon_N\left(\Pi^{\mathcal{L}}\right) - \text{Regret}_N^{\mathcal{L}}\right]. \tag{24}$$

Now let $\Pi_m^{\mathcal{E}} = \Pi^\circ \cup \pi_m$. Following the same reasoning strategy as above, we will have lower bound for RPI as $\mathbb{E}_{s \sim d_0}[f_m^+(s)] + \mathbb{E}\left[\Delta_N - \epsilon_N\left(\Pi^{\mathcal{L}}\right) - \text{Regret}_N^{\mathcal{L}}\right]$. Since $\mathbb{E}_{s \sim d_0}[f_m^+(s)] \geq \mathbb{E}_{s \sim d_0}[f_0^+(s)]$, we have performance lower bound of RPI no worse than MAMBA.

**Remark.** MAPS (Liu et al., 2023) retains MAMBA's lower bound but enhances sample efficiency and reduces the bias in $\text{Regret}_N^{\mathcal{L}}$. The inherent uncertainty of the optimal policy $\pi^\star \in \Pi_m^{\mathcal{E}}$ makes an unbiased $f^+$ estimate challenging. The regret term $\text{Regret}_N^{\mathcal{L}}$ is bounded by:

$$\mathbb{E}\left[\text{Regret}_N^{\mathcal{L}}\right] \leq O\left(\left(\beta^+ + \beta^\epsilon\right) N + \sqrt{vN}\right),$$

where $\beta^+$ is the estimation bias that results from selecting the non-optimal policy $\hat{\pi}^\star$ in $\Pi_m^{\mathcal{E}}$ for a given state, and $\beta^\epsilon$ is the value estimation error w.r.t. the true value for given state of selected policy $\hat{\pi}^\star$ and $v$ represents the variance term.

MAPS improves upon MAMBA's sample complexity, reducing bias in its regret bound via an active policy selection mechanism. Our work builds on MAPS, emphasizing empirical enhancements in active policy selection with the integration of the learner policy in $\Pi_m^{\mathcal{E}}$.

## D Experimental Details

### D.1 Baselines

**AggreVaTeD**  AggreVaTeD (Sun et al., 2017) is a differentiable version of AggreVaTe, which focuses on a single oracle scenario. AggreVaTeD allows us to train policies with efficient gradient update procedures. AggreVaTeD models the policy as a deep neural network and trains the policy using differentiable imitation learning. By applying differentiable imitation learning, it minimize the difference between the expert's demonstration and the learner policy behavior. AggreVaTeD learns from the expert's demonstration while interact with the environment to outperform the expert.

**Max-Aggregation**  We have developed a variant of the Max-aggregation policy as outlined in Equation (16) that is specifically designed for pure imitation learning using multiple oracle sets. When utilizing a single oracle, it effectively reduces to AggreVateD. Our approach builds on the existing MAMBA framework by setting the lambda value in the loss function to zero. While max-aggregation may not always yield the optimal policy, it offers the advantage of being able to achieve results with fewer samples, making it a more sample-efficient option.

**LOKI-variant**  LOKI (Cheng et al., 2018) is strategy for policy learning that combines the imitation learning and reinforcement learning objectives in a two-stage manner for the single oracle setting. In the first stage, LOKI performs imitation learning for a small but random number of iterations and then switches to policy gradient reinforcement learning method for the second stage. LOKI is able to outperform a sub-optimal expert and converge faster than running policy gradient from scratch. In this work, we propose a variation of LOKI that adapts to multiple experts. During the first-half of training (i.e., the first stage) we perform Max-aggregation style imitation learning, and then perform pure reinforcement learning as the second stage.

**PPO-GAE**  Schulman et al. (2015) proposed the generalized advantage estimator (GAE) as a means of solving high-dimensional continuous control problems using reinforcement learning. GAE is used to estimate the advantage function for updating the policy. The advantage function measures how much better a particular action is compared to the average action. Estimating the advantage function with accuracy in high-dimensional continuous control problems is challenging. In this work, we propose PPO-GAE, which combines PPO's policy gradient method with GAE's advantage function estimate, which is based on a linear combination of value function estimates. By combining the advantage of PPO and GAE, PPO-GAE (Schulman et al., 2017) achieved both sample efficiency and stability in high-dimensional continuous control problems.

**MAMBA**  MAMBA (Cheng et al., 2020) is the SOTA work of learning from multiple oracles. It utilizes a mixture of imitation learning and reinforcement learning to learn a policy that is able to imitate the behavior of multiple experts. MAMBA is also considered as interactive imitation learning algorithm, it imitates the expert and interact with environment to improve the performance. MAMBA randomly select the state as switch point between learner policy and oracle. Then, it randomly selects the oracle to roll out. It effectively combines the strengths of multiple experts and able to handle the case of conflicting expert demonstrations.

**MAPS**  MAPS (Liu et al., 2023) is a policy improvement algorithm that performs imitation learning from multiple suboptimal oracles. It actively chooses the oracle to imitate based on their value function estimates and identifies the states that require exploration. By introducing two variations, Active Policy Selection (APS) and Active State Exploration (ASE), MAPS improves the sample efficiency of MAMBA. The MAPS variant selects the most promising oracle, denoted as $k_\star$, for rollout, utilizing the resulting trajectory to refine the value function estimate $\hat{V}^{k_\star}(s_t)$. This approach aims to minimize the chances of selecting an inferior oracle for a given state $s_t$, thereby reducing both the sample complexity and gradient estimation bias. On the other hand, the ASE variant of MAPS deliberates whether to continue with the current policy or switch to what is believed to be the most promising oracle, similar to APS, by leveraging an uncertainty measure over the current state. In this study, we adopt MAPS variant as our baseline method.

### D.2  GYM ENVIRONMENTS

We evaluate RPI and compare its performance to the aforementioned baselines on the Cheetah-run, CartPole-swingup, Pendulum-swingup, and Walker-walk tasks from the DeepMind Control Suite (Tassa et al., 2018) and Window-close, Faucet-open, Drawer-close and Button-press from Meta-World (Yu et al., 2020). In addition, we conduct experiments on a modified sparse reward Meta-World environment, which is considered to be a more challenge task.

### D.3  SETUP

**Setup.**  In order to ensure a fair evaluation, all baselines are assessed using an equal number of environment interaction steps (training steps). Each training iteration involved a policy rollout for the same number of steps. We note that there is a discrepancy in the amount of data available to the learners of RPI and PPO-GAE. RPI (MAPS, MAMBA, Max-Aggregation) uses some of the interactions to learn the value function for each Oracle, which results in relatively less data for its learner, whereas PPO-GAE utilizes all the environment interactions to update all benefits for its learner policy. Thus, in this work, we balance the transition buffer size for each algorithm to make them have approximately same number of stored transitions for learner policy improvement. We average the result based on 5-10 trials.

### D.4  IMPLEMENTATION DETAILS OF RPI

We provide the details of RPI in Algorithm 1 as Algorithm 2. Algorithm 2 closely follows Algorithm 1 with a few modifications as follows:

- In line 5, we use a buffer with a fixed size ($|\mathcal{D}_n| = 19,200$) for each oracle, and discard the oldest data when it fills up.

---

**Algorithm 2** Robust Policy Improvement (RPI)

---

**Input:** Learner policy $\pi_1$, oracle set $\Pi = \left\{\pi^k\right\}_{k \in [K]}$, function approximators $\{\hat{V}^k\}_{k \in [K]}$, $\hat{V}_n$

**Output:** The best policy in $\{\pi_1, ..., \pi_N\}$.

1: **for** $n = 1, 2, \ldots, N - 1$ **do**
2:     Construct an extended oracle set $\Pi^{\mathcal{E}} = \left[\pi^1, \pi^2, \ldots, \pi^k, \pi_n\right]_{k \in [|\Pi|]}$.
3:     Sample $t_e \in [H-1]$ uniformly random. We have a buffer with a fixed size ($|\mathcal{D}_n| = 19,200$) for each oracle, and we discard the oldest data when it fills up.
4:     Switch to $\pi^{k_\star}$
5:     Roll-in $\pi_n$ up to $t_e$, select $k_\star$ (10), and roll out $\pi^{k_\star}$ to collect data $\mathcal{D}^k$.
6:     Update $\hat{V}^{k_\star}$ using $\mathcal{D}^k$.
7:     Roll-out the learner policy $\pi_n$ for a specified steps $(2,048)$, and add them to the buffer $\mathcal{D}'_n$.
8:     Update $\hat{V}_n$ using $\mathcal{D}'_n$.
9:     Compute advantage $\hat{A}^{\text{GAE+}}$ (11) and gradient estimate $\hat{g}_n$ (14) using $\mathcal{D}'_n$.
10:     Perform PPO style policy update on policy $\pi_n$ to $\pi_{n+1}$.

---

- In line 7, we roll-out the learner policy until the buffer reaches a fixed size ($|\mathcal{D}'_n| = 2,048$), and then empty it once we use the roll outs to update the learner policy. This stabilizes the training compared to storing a fixed number of trajectories in the buffer, as MAMBA does.

- In line 10, we use PPO with a $\max^+$ actor-critic style policy update.

- We pretrain the value function $\hat{V}^k$ of oracle $k$ before the main training loop, with trajectories generated by rolling out $\pi^k$ from the initial states. In the main training loop, we train $\hat{V}^k$ using the corresponding rolled-out trajectories, bootstrapped only by itself. This is the same strategy as in MAMBA and MAPS. Similarly, we train the learner value function $\hat{V}_n$ using only the trajectories rolled-in with $\pi_n$, bootstrapped only by itself.

### D.5    Hyperparameters and architectures

Table 3 provides a list of hyperparameter settings we used for our experiments. We use an ensemble of MLPs to predict an oracle's value. With five identical MLPs that are separately initialized at random, we train each MLP separately for the same dataset. At inference time, we collect predictions from all MLPs for a single input and compute the mean and standard deviation.

### D.6    Computing infrastructure and wall-time comparison

We conducted our experiments on a cluster that includes CPU nodes (approximately 280 cores) and GPU nodes (approximately 110 Nvidia GPUs, ranging from Titan X to A6000, set up mostly in 4- and 8-GPU configurations). Based on the computing infrastructure, we obtained the wall-time comparison in Table 4 as follows.

## E    Supplemental Experiments

### E.1    Comparing RPI against baselines with a data advantage

In the main paper, we followed an experimental setup where we assumed that the learner had access to approximately the same quantity of transitions, while lacking access to the oracle's offline trajectory. However, some of the baseline algorithms, such as MAPS (Liu et al., 2023) were originally evaluated under a different setting in the literature: They assume that the learner can access additional data from the oracles' pre-trained offline dataset. In this section, we run MAPS under such a setting, in order to provide more comprehensive evaluation that is consistent with the literature. Note that under this experimental setup, MAPS has approximately twice the amount of data compared to RPI. We refer to this variant as MAPS-ALL.

In contrast to MAPS in our original configuration, the performance of MAPS-ALL doubles in the Cheetah environment (as shown in Figure 5(a)) and the Pendulum environment (as depicted in Figure

| Parameter | Value |
|---|---|
| **Shared** | |
| Learning rate | $3 \times 10^{-4}$ |
| Optimizer | Adam |
| Nonlinearity | ReLU |
| # of functions in a value function ensemble | 5 |
| **Oracle** | |
| # of oracles in the oracle set ($K$) | 3 |
| The buffer size for oracle $k$ $\left( \lvert \mathcal{D}^k \rvert \right)$ | 19200 |
| # of episodes to rollout the oracles for value function pretraining | 8 |
| **Learner** | |
| Horizon of MetaWorld and DMControl ($H$) | 300, 1000 |
| Replay buffer size for the learner policy $\left( \lvert \mathcal{D}'_n \rvert \right)$ | 2048 |
| GAE gamma ($\gamma$) | 0.995 |
| GAE lambda ($\lambda$) for AggreVaTeD and Max-Aggregation
for LOKi-variant
for RPI | 0
0 or 1
0.9 |
| # of training steps (rounds) ($N$) | 100 |
| # of episodes to perform RIRO (Alg 1, line 4) per training iteration | 4 |
| mini-batch size | 128 |
| # of epochs to perform gradient updates per training iteration | 4 |

Table 3: RPI Hyperparameters.

| Methods | Cheetah | Cartpole | Walk-Walker | Pendulum |
|---|---|---|---|---|
| **MAPS** | 1h 18m | 1h 10m | 1h 41m | 1h 17m |
| **MAMBA** | 1h 23m | 59m | 2h 14m | 1h 21m |
| **LOKI-variant** | 1h 17m | 1h 36m | 2h 11m | 1h 12m |
| **PPO-GAE** | 54m | 58m | 1h 10m | 49m |
| **MAX-aggregation** | 1h 5m | 1h 34m | 2h 25m | 1h 13m |
| **RPI** | 57m | 58m | 1h 43m | 1h 18m |

Table 4: Wall-time comparison between different methods.

5(c)). Moreover, the performance of MAPS-ALL surges between middle and end of rounds in the Pendulumn environment. This behavior mirrors what was reported in the original MAPS paper as well. In the Walker-walk environment (as illustrated in Figure 5(b)), MAPS-ALL demonstrates an approximate 10% improvement. For the Cartpole environment (Figure 5(d)), MAPS-ALL's performance increases by around 20%. MAPS-ALL exhibits overall performance similar to that of its original paper, with any differences caused by the difference in the oracle set. Notably, as a result, MAPS-ALL distinctly outperforms RPI only in the Pendulum environment. RPI's performance remains comparable to MAPS-ALL in the Cheetah environment and significantly surpasses the MAPS-ALL baseline in the Walker-Walker and Cartpole environments, despite utilizing much less data.

## E.2 META-WORLD EXPERIMENTS (DENSE REWARD)

In Fig. 6, we conducted additional experiments comparing RPI and state-of-the-art (SOTA) methods {MAMBA, MAPS}, as well as the best-performing oracle and PPO-GAE, across the Meta-World benchmarks. The tasks are including (1) `window-close`, (2) `faucet-open`, (3) `drawer-close`, and (4) `button-press`. RPI demonstrates superior performance compared to all baselines in the majority of environments, with the exception of the button-press task.

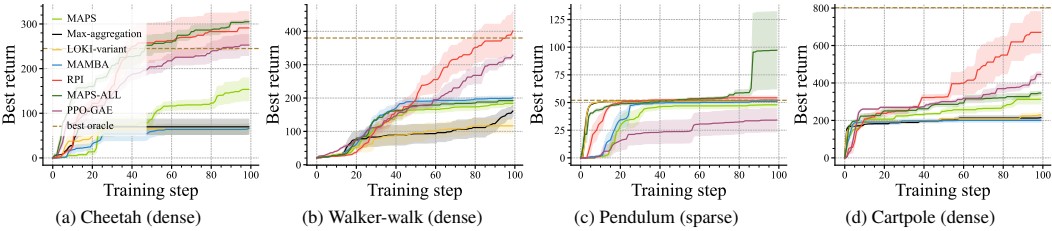

Figure 5: Running MAPS in the original paper's setting.

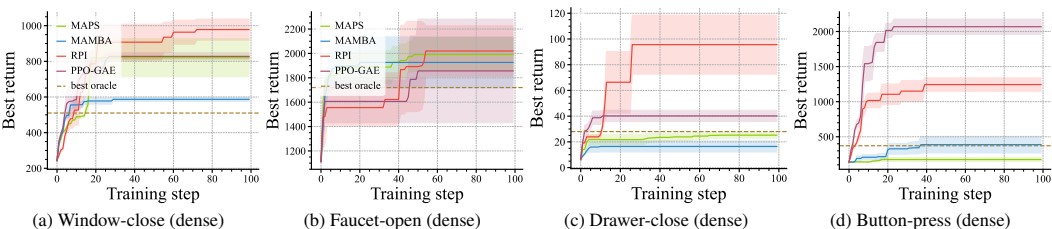

Figure 6: Experimental results on the Meta-World benchmark with dense reward.

### E.3 META-WORLD EXPERIMENTS (SPARSE REWARD)

In Fig. 7, to further demonstrate the advantages of imitation learning, we modified the Meta-World environment to create a more challenging sparse reward environment. In this environment, the agent only receives a reward of 1 upon success; otherwise, it receives a reward of 0. We then compared the performance of the RPI and state-of-the-art (SOTA) imitation learning-based methods MAMBA, MAPS, as well as the pure RL method PPO-GAE, and the best-performing oracle across the Meta-World benchmarks. The tasks include (1) `window-close`, (2) `faucet-open`, (3) `drawer-close`, and (4) `button-press`. In these sparse reward environments, when provided with a good oracle, the imitation learning-based approach demonstrates its advantage over the pure RL approach. RPI, MAPS, and Mamba outperform PPO-GAE by a factor of 3 in the `button-press` environment. When provided with a bad oracle, RPI can still outperform MAPS and MAMBA in the `faucet-open` environment. Moreover, even with a poor oracle, RPI outperforms both IL-based approaches (MAMBA, MAPS) and the RL-based approach (PPO-GAE) in the `window-close` environment, showcasing that RPI enjoys benefits from both RL and IL aspects.

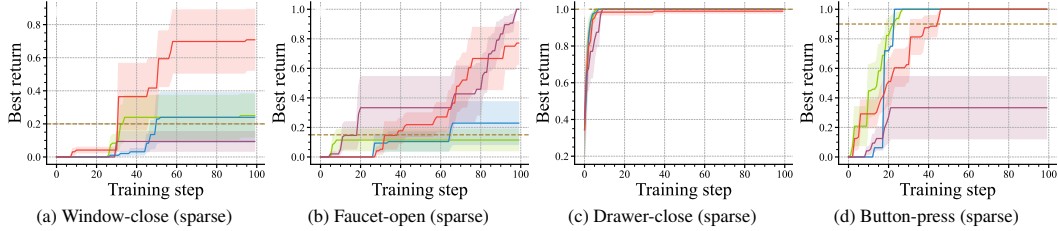

Figure 7: Experimental results on the Meta-World benchmark with sparse reward.

### E.4 SINGLE/EMPTY ORACLE SET

In Fig. 2, we mainly discuss experiment on multiple oracle set. In Fig. 8, we demonstrate that RPI is also robust enough to handle single or empty oracle set.

**Single oracle setting.** In Fig. 8(b), we demonstrate that RPI outperforms all other baselines in a single oracle setting as well. This is consistent with the results observed in the multiple-experts setting. Fig. 8(c) demonstrates that providing an oracle with mediocre performance to RPI boosts the performance rather than providing a near-optimal oracle. Since we train oracles' value functions from the oracle rollouts, the value function of the near-optimal oracle may not have seen the "bad" states

that the learner policy would encounter in the early stage. This leads to inaccuracy in the predicted values for such states. In comparison, the value function for a mediocre oracle would be able to produce accurate predictions on such states.

**No oracle environment.** When there are no experts available, the performance of imitation learning-based approaches will inevitably degrade. However, as shown in Fig. 8(a), RPI can adapt to such a scenario by regressing to pure reinforcement learning. Since we extend RPI based on PPO-GAE, it achieves a similar level of performance to PPO-GAE when the oracle set is empty.

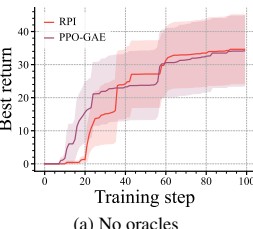
(a) No oracles

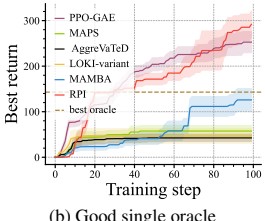
(b) Good single oracle

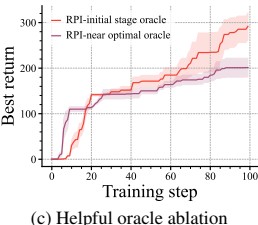
(c) Helpful oracle ablation

Figure 8: **Ablation study.** (a) Comparing RPI and PPO-GAE under no oracle Pendulum environment. (b) Comparing RPI and six baseline under single expert Cheetah environment. (c) ablation study on oracle quality under Cheetah environment.

### E.5 ABLATION ON CONFIDENCE THRESHOLD $\Gamma_s$

In Table 5, we perform an ablation study of threshold $\Gamma_s$ in Eqn. 12 of robust policy gradient component. We treat $\Gamma_s$ as a hyperparameter the choice of which depends on the user's risk aversion. Empirically, we find that a setting of $\Gamma_s = 0.5$ robustly works well in nearly every setting in our experiments across Deepmind Control suite, Meta-World (Dense reward), and Meta-world (Sparse reward) environment, the one exception being the Pendulum domain. To better understand how the choice of $\Gamma_s$ effects overall performance, we conducted a set of experiments in which we ran RPI on the Deepmind Control Suite using different values for $\Gamma_s \in [0, 0.5, 1, 3, 5]$. As the following table shows, setting $\Gamma_s = 0.5$ yields the best performance for all but the Pendulum environment.

In practice, one can first use roll outs of each oracle to estimate the standard deviation and associated confidence intervals of their ensemble values. A conservative user could then start by setting $\Gamma_s$ based on a probabilistic lower-bound of $\sigma$ and subsequently tune the hyperparameter according to user's risk aversion preference.

| Environment | Round | $\Gamma_s = 0$ | $\Gamma_s = 0.5$ | $\Gamma_s = 1$ | $\Gamma_s = 3$ | $\Gamma_s = 5$ |
|---|---|---|---|---|---|---|
| **Cheetah** | 100 | 252.7 ± 23.2 | **291.2** ± 36.3 | 251.4 ± 15.1 | 53.4 ± 20.0 | 81.3 ± 20.8 |
| **Walker-walk** | 100 | 328.7 ± 6.5 | **402.2** ± 57.7 | 253.0 ± 43.5 | 31.8 ± 1.4 | 38.2 ± 1.9 |
| **Pendulum** | 100 | 34.2 ± 23.5 | 38.0 ± 10.4 | 45.6 ± 2.3 | **54.3** ± 1.5 | 52.1 ± 0.1 |
| **Cartpole** | 100 | 445.7 ± 13.5 | **670.4** ± 110.1 | 394.8 ± 50.6 | 301.7 ± 60.0 | 303.2 ± 4.0 |

Table 5: Tuning the confidence threshold $\Gamma_s$.

### E.6 ABLATION ON UCB/LCB POLICY SELECTION

| Environment | Round | RPI-RAPS(LCB/UCB) | RPI-MEAN |
|---|---|---|---|
| **Cheetah** | 100 | **291.2** ± 36.3 | 263.0 ± 33.7 |
| **Walker-walk** | 100 | **402.2** ± 57.7 | 342.7 ± 18.8 |
| **Pendulum** | 100 | **54.3** ± 1.5 | 53.8 ± 0.5 |
| **Cartpole** | 100 | **670.4** ± 110.1 | 354.3 ± 65.2 |
| **Overall** | 100 | **1418.1** | 1013.8 |

Table 6: Ablation study on confidence aware UCB/LCB vs MEAN policy selection

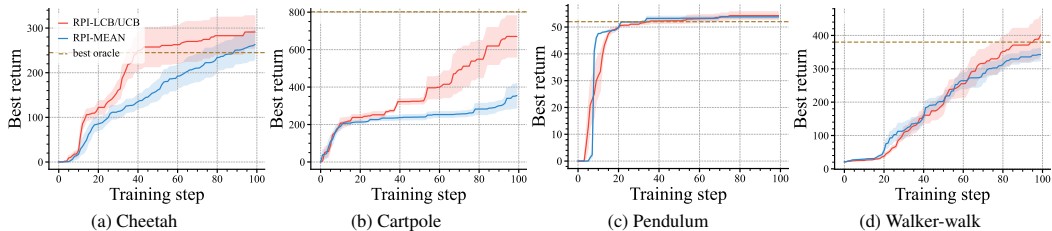

Figure 9: Experimental results on ablation study on confidence aware UCB/LCB vs MEAN policy selection.

We conducted an ablation study that compares RPI-LCB/UCB, which takes uncertainty into account as follows:

$$K = \arg\max \left( \overline{\hat{V}^1(s)}, \overline{\hat{V}^2(s)}, \ldots, \underline{\hat{V}^{K+1}(s)} \right)$$

against RPI-MEAN, which does not consider uncertainty as:

$$K = \arg\max \left( \hat{V}^1(s), \hat{V}^2(s), \ldots, \hat{V}^{K+1}(s) \right).$$

The experimental results presented in Fig. 9 and Table 6 demonstrate that the RPI-LCB/UCB strategy outperforms RPI-MEAN across all benchmarks by an *overall* margin of 40%. This highlights the significance of incorporating uncertainty in the policy selection strategy.

