# OpenReview forum: "Blending Imitation and Reinforcement Learning for Robust Policy Improvement"
_ICLR.cc/2024/Conference — ICLR 2024 spotlight_

### Official Review · Reviewer_EG6V · 2023-10-22

**Soundness:** 3 good
**Presentation:** 3 good
**Contribution:** 3 good
**Rating:** 8
**Confidence:** 3

**Summary:**

This paper proposes a method that combines RL and IL to learn a policy that improves upon a set of imperfect oracles. The method extends the MAMBA algorithm [1] by incorporating the learner policy into the set of oracles and using uncertainty-aware value-estimates to define the exploration strategy and advantage estimator. These changes enable the proposed method to behave like an RL algorithm in cases where the oracles do not provide beneficial supervision. The proposed method achieves good results in mujoco and meta-world tasks, beating most of the oracle policies and baselines.

**Strengths:**

- The proposed method beats a range of relevant baselines in a diverse set of benchmarks.
- The sections 1 to 5 are generally well written and reasonably easy to follow. Caveat: I did not verify the derivations myself.

**Weaknesses:**

## Presentation
- The method description has missing details. These are also issues with soundness of the paper.
    - The necessary details for understanding equation (12) are not clearly presented. See questions.
    - The value ensemble is central to the proposed method, yet there are only two sentences at the bottom of page 5 discussing how the value ensemble is trained.
- The theory section borrows from MAMBA, which is only natural since these methods are closely related, but it would have been clearer to emphasize the points of departure from MAMBA more and perhaps move the parts that are more closely shared to the appendix.
- Nitpick:
    - The definition of policy gradient with advantage is cited for Schulman, but there are prior works considering policy gradients with advantage estimation, e.g., [2].
    - The abstract mentions real-world scenarios but the paper does not talk about them.



## Soundness
- Computing the update requires evaluating the value ensembles for all of the oracles. Compared to normal RL algorithm, e.g., PPO, this means number of oracles times the size of ensemble more forward passes. Depending on the number of oracles and ensemble members, this may equate to a lot more expensive loss evaluation. In addition to the missing details about the ensemble size and number of oracles, it would be good to include some wall-time comparisons between the different methods.

## Experimental evaluation
- Cartpole is a bit too easy environment for the otherwise well-executed ablation study.
- In the appendix, it says that the value ensembles are pre-trained before the main loop starts. That sounds like the proposed method uses more samples (and more compute) than the baselines. This should be mentioned in the main paper and discussed as a limitation.

## Summary
The proposed method seems to do well, but the paper is missing too many details and is too hard to follow at parts that it is hard to evaluate whether the contribution is impactful or not. I think including a thorough discussion about the relationship of the proposed method and MAMBA as well as limitations would go a long way to make this easier to understand. Additionally, adding all of the missing details that I have listed here and in the questions would help.

**Questions:**

- Equation (12)
    - What does equation (12) mean when the "when"-clause equates to false?
    - What does $\hat{V}_{\mu}^{|\Pi^{\varepsilon}|}$ mean?
- How many members are there in the value-function ensemble?
- How many oracles does each experiment have?
- Why are the replay buffers for the oracles so small? The oracle policies do not change, so the data does not become more off-policy as the training proceeds.
- What are the x-axes in the result figures? Surely the algorithms do not converge in 100 gradient steps?
- How many timesteps are sampled for value-function pre-training? How would the learning curves look like if these steps were included on the x-axis? In general, using environment steps as the x-axis would make it easier to compare methods that are otherwise so different.


[1] Policy Improvement via Imitation of Multiple Oracles, Cheng et al. 2020

[2] Policy Gradient Methods for Reinforcement Learning with Function Approximation, Sutton et al. 2000

---

> ### Author Response · Authors · 2023-11-17
> **Thanks for your review!**
>
> We greatly appreciate your comprehensive review of our manuscript. Below, we address each of your questions in detail:
>
>
> > ***Q1:*** "I think including a thorough discussion about the relationship of the proposed method and MAMBA as well as limitations would go a long way to make this easier to understand."
>
> ***A:*** Thank you for your advice! We have added a detailed paragraph in Appendix A.1 (additional notes on related work) where we discuss the relationship between the proposed method and MAMBA, as well as their limitations (see Appendix B.2 limitations of the prior art). Please see the updated PDF. In the meantime, we created the following table in hopes that it addresses your questions/concerns.
>
>
> |         | RPI    | MAPS [ICML'23] | MAMBA [Neurips'20] |
> |--------      |--------     |--------       |  --------   |
> | Objective| Perform active IL and RL for robust policy improvement from multiple suboptimal oracles. Addressed the limitation of MAPS and MAMBA with regard to the robustness.  | Improve the sample efficiency of MAMBA by active policy selection (For MAP-SE, it also proposed active state exploration)| Perform online imitation learning from multiple suboptimal oracles. |
> | Contribution | (1) Proposed the $\text{max}^+$ framework for robust learning (2) Proposed $\text{max}^+$-aggregation baseline  (3) Robust policy improvement (RPI) algorithm (4) Robust Active Policy Selection strategy (RAPS)  (5) Robust Policy Gradient (RPG) (6) Provided theoretical analysis. | MAPS (1) Active policy selection (APS) (2) Provided theoretical analysis. | (1) Proposed max-aggregation baseline. (2) Novel algorithm MAMBA, max-aggregated baseline. (3) Regret-based performance guarantee for MAMBA.
> | Experiments |(1) Cheetah (Dense) (2) Walker-walk (Dense) (3) Cartpole (Dense) (4) Pendulum (Sparse) (5) Faucet-open (Dense) (6) Faucet-open (Sparse) (7) Drawer-close (Dense) (8) Drawer-close (Sparse) (9) Button-press (Sparse) (10)Button-press (Dense) (11) Window-close (Sparse) (12) Window-close (Dense)| (1)Cheetah (Dense)  (2)Walker-walk (Dense) (3)Cartpole (Dense) (4)Pendulum (Sparse)| (1) DIP (Dense) (2) Cheetah (Dense) (3) Ant (Dense) (4) Cartpole  (Dense)|
> | Baseline  | (1)MAPS (2)MAMBA (3)Max-aggregation (4)LOKI-variant (5)PPO-GAE (6)Best oracle| (1)MAMBA (2)PPO-GAE (3)Best oracle | (1)PG-GAE (2)Best oracle|
> | Limitation  |    lacking robustness guarantee against the optimal pure RL policy (a limitation shared by MAPS/MAMBA, although RPI demonstrates superior empirically performance).   | Not robust to the oracle quality (such as pure adversarial oracle setting)   | (1) No active oracle selection (Sample inefficiency)  (2) Not robust to the oracle quality (such as pure adversarial oracle setting) |
>
>
>
> >  ***Q2:*** "...include some wall-time comparisons between the different methods.."
>
> ***A:*** Thank you for the suggestion! We have added Table 4 in Appendix D.6 "Computing Infrastructure and Wall-Time Comparison" that provides the wall-time comparison that you requested.
>
> > ***Q3:*** "In the appendix, it says that the value ensembles are pre-trained before the main loop starts. That sounds like the proposed method uses more samples (and more compute) than the baselines. This should be mentioned in the main paper and discussed as a limitation."
>
> ***A:*** Thank you for pointing this out. You are correct that the IL methods (including RPI) use slightly more samples compared to PPO-GAE as a result of pretraining the value function of the oracles. However, we emphasize that we use the same number of samples for RPI as we do for the IL+RL baselines (i.e., MAMBA, MAPS, Max-Aggregation, and LOKI) to ensure a fair comparison. We updated Section 7 (Baselines) to make this clear.
>
>
> >  ***Q4:*** "The theory section borrows from MAMBA, which is only natural since these methods are closely related, but it would have been clearer to emphasize the points of departure from MAMBA more and perhaps move the parts that are more closely shared to the appendix?"
>
> ***A:*** We appreciate your feedback! We agree with the reviewer that there are similarities with MAMBA and that the paper can be clearer regarding how RPI differs. Despite these similarities, we respectfully note that our theory is not directly borrowed from MAMBA. Different from MAMBA, our approach considers an oracle set that includes the learner policy, which we call *extended oracle set*. With this change, the (extended) oracle set that the algorithm work with is not static anymore (due to the improvement of learner's policy). This changes the theoretical analysis in a non-trivial manner. While Proposition 5.1 is marked as "adapted from MAMBA," our bound differs from MAMBA's for this reason. We updated the main paper to make this departure from MAMBA clear.

---

> > ### Author Response · Authors · 2023-11-17
> >
> > > ***Q5:*** "The definition of policy gradient with advantage is cited for Schulman, but there are prior works considering policy gradients with advantage estimation"
> >
> > ***A:*** Thank you for catching this! We added the following references to the paper.
> >
> > Sutton et al., Policy Gradient Methods for Reinforcement Learning with Function Approximation. JMLR, 1999
> >
> > Greensmith et al., Variance Reduction Techniques for Gradient Estimates in Reinforcement Learning. NeurIPS, 2004
> >
> > >  ***Q6:*** "What does equation (12) mean when the "when"-clause equates to false?"
> >
> > ***A:*** Thanks for asking this question. We have updated the Eq. 12 in the revised version to address this confusion. Essentially, we first compute $k^* = argmax_{k\in[{|\Pi^E|}]}\hat{V}^k_{\mu}(s)$ to get the index $k^*$ of the policy with the highest value. If the estimated standard deviation for the value ($\sigma_{k^*}(s)$) is larger than a threshold, we use the estimated value for the learner policy as $\hat{f}^+(s)$. If not we use $\hat{f}^+(s) = \hat{V}^{k^*}_{\mu}(s)$.
> >
> > > ***Q7:*** "What does $\hat{V}_{\mu}^{|\Pi^E|}$ mean?"
> >
> > ***A:***  $\hat{V}_{\mu}^{|\Pi^E|}$ refers to the estimated value function of the learner policy, where $|\Pi^E|$ denotes the index of the extended policy, which is the learner policy.
> >
> > To make it clearer, we update it to $\hat{V}_{\mu}^{\pi_n}$. Please check Eq. 12 in revised version for details.
> >
> > > ***Q8:*** "How many members are there in the value-function ensemble?"
> >
> > ***A:*** We followed the state-of-the-art MAPS for value-function ensemble numbers and set it to five. Please check the Appendix D.5 Hyperparameters and architectures for more details.
> >
> > > ***Q9:*** "The value ensemble is central to the proposed method, yet there are only two sentences at the bottom of page 5 discussing how the value ensemble is trained."
> >
> > ***A:*** Thanks for asking the question. We add a more detail about the value ensemble network including how it is trained in the Appendix D.5 Hyperparameter and architectures. Please check our updated PDF file for more details.
> >
> >
> >
> > > ***Q10:*** "How many oracles does each experiment have?"
> >
> > ***A:*** In each experiment, we have three diversified oracles. Please see the updated Table 3 in Appendix D.5 and the caption of Figure 2.
> >
> > > ***Q11:*** "Why are the replay buffers for the oracles so small? The oracle policies do not change, so the data does not become more off-policy as the training proceeds."
> >
> > ***A:*** We set the size of replay buffer to $19,200$. Indeed, the oracle policies are fixed and increasing replay buffer seem to have a positive effect, however, in our preliminary experiments (not reported on the paper) we figured that further increasing the replay buffer does not affect the performance.
> >
> > > ***Q12:*** "How many timesteps are sampled for value-function pre-training? What would the learning curves look like if these steps were included on the x-axis? In general, using environment steps as the x-axis would make it easier to compare methods that are otherwise so different."
> >
> > ***A:*** In order to pretrain a value function for each oracle, we collected 8 episodes (it amounts to 8k steps for DMControl, 2.4k steps for MetaWorld) per oracle. For each training step of the RPI algorithm, the environment is queried for 4 episodes (roll-in + roll-out), followed by 2,048 steps (rolling out the learner's policy).
> >
> >
> > > ***Q13:*** "What are the x-axes in the result figures? Surely the algorithms do not converge in 100 gradient steps?"
> >
> > ***A:*** The x-axis in the results figures represent the training steps of our RPI algorithm. Answering your question concisely, each training step indeed contains many gradient updates. To elaborate on this, we will give the implementation details for gradient updates (to update the learner's policy): 1. We first prepare a set of transitions. This comes from the roll-in (see Alg. 1, line 4) and full roll-out (see Alg. 1, line 6) of the learner policy; 2. We then create a set of mini-batches (of size 128) by grouping the transitions; 3. We iteratively perform gradient updates on the mini-batches for 2 epochs (i.e., going over all of the mini-batches twice). This entire process makes up a single training step. Please feel free to let us know if there are other details that should be elaborated.
> >
> > ---
> > We hope that our responses address your remaining concerns and provide clarification on the contributions and methodologies of our work. If you have any further inquiries, please kindly let us know.
> > Thank you!

---

> > > ### Comment · Reviewer_EG6V · 2023-11-21
> > >
> > > Thanks for the detailed response. I have read it as well as the other reviews and responses. Most of my questions have been answered adequately. I raise my score to 6.
> > >
> > > I would still like to see improvements w.r.t. Q4 per your labeling. I believe this is the same issue as 6RE6's latest comment about the similarity of section 4 to MAMBA. As 6RE6 states, section 4 extends the definitions in MAMBA in a straightforward way. Including a clear discussion on how the definitions relate to corresponding ones MAMBA would be good.

---

> ### Author Response · Authors · 2023-11-22
> **Thank you for your additional feedback!**
>
> Thank you for your additional feedback! We are glad that our initial responses adequately answered most of your questions. We are also pleased that it has convinced you to change your score. We would like to take the opportunity to further address your remaining question.
>
> > Q4: "The theory section borrows from MAMBA, which is only natural since these methods are closely related, but it would have been clearer to emphasize the points of departure from MAMBA more and perhaps move the parts that are more closely shared to the appendix?"
> >
> > I would still like to see improvements w.r.t. Q4 per your labeling. I believe this is the same issue as 6RE6's latest comment about the similarity of section 4 to MAMBA. As 6RE6 states, section 4 extends the definitions in MAMBA in a straightforward way. Including a clear discussion on how the definitions relate to corresponding ones MAMBA would be good.
>
>
> ***A:*** We agree that section 4 has similarity to MAMBA, as both share the same problem setting. The main difference arises from lifting the static oracle set into our dynamic extended oracle set. However, to formally state the problem and establish our algorithm and analysis, we find it necessary to rigorously set up the notations and keep the work self-contained, which unavoidably overlaps with MAPS/MAMBA as a special case. In fact, many of the notations also inherit from MAMBA due to the shared problem setting, and we have moved most of the background discussion to `Appendix B1` (“Additional algorithms for learning from multiple oracles”) for clarity. In addition, to make it clearer, we added a footnote in section 4 where we explicitly state, "When we exclude the learner’s policy from the extended oracle set, this reduces to the max-aggregation policy, which was used in MAMBA (Cheng et al., 2020)" to elucidate the connection with existing work (Please find details in the revised version.). Starting from Section 5, our approach clearly departs from MAMBA as it really focus on selecting between learner's policy and oracle policies, as well as the robust update in policy gradient.
>
> ---
>
> We hope our responses have addressed your remaining questions. Again, we greatly appreciate your valuable time and input!

---

> > ### Comment · Reviewer_EG6V · 2023-11-23
> >
> > Thanks for the continued engagement. I think the similarities to MAMBA are handled much more clearly in the newest revision. I raise my score to 8.

---

### Official Review · Reviewer_oW5n · 2023-10-29

**Soundness:** 4 excellent
**Presentation:** 4 excellent
**Contribution:** 4 excellent
**Rating:** 8
**Confidence:** 2

**Summary:**

This paper proposes the max+ algorithm, which is a hybrid imitation + reinforcement learning algorithm which utilizes a set of expert policies to aid a policy in initial learning and exploration, and gradually transitions to reinforcement learning as the learner policy improves. The method maintains a value estimate for each expert policy as well as the learner's, and uses the value functions to construct an update rule that improves over the best expert in the set. The authors then apply their proposed method to the continuous control domain on DM Control Suite and Metaworld tasks, where they demonstrate that their method achieves greater learning speed than prior baselines.

**Strengths:**

- The paper is clear and well written. The authors have done a good job of presenting an overview of their method, giving theoretical justification of why the method works, and benchmarking against several state-of-the-art baselines.
- The experiments are detailed, and I appreciate the ablation studies which show the usefulness of the robust policy selection rule vs naive rule, and the transition from utilizing the expert policies early in training while transitioning smoothly to RL as the policy improves.
- I am not familiar enough with the literature in this subdomain to judge the degree of novelty in the approach, but based on the information presented in the paper the proposed update rule (the robust policy gradient, and robust policy selection) is novel.

**Weaknesses:**

- For reproducibility purposes, it would be great if the authors could include a table of algorithm hyperparameters used in the appendix, and hyperparameter sweeping strategies.

**Questions:**

- How would the following naive baseline perform? Initialize from the expert policy, and then finetune from there with RL. It seems like this baseline could perform quite well in the proposed experimental setup, given that PPO-GAE is one of the baselines and performs quite well.

---

> ### Author Response · Authors · 2023-11-17
> **Thanks for your review!**
>
> We appreciate your in-depth review of our work! Below please find our detailed responses to your questions below.
>
> > ***Q:*** "For reproducibility purposes,it would be great if the authors could include a table of algorithm hyperparameters used in the appendix..."
>
> ***A:*** Thanks for the recommendation! In the revised version, we added a table that summarizes the hyper-parameter settings in Appendix D.5. Please check our updated version file for more details.
>
> > ***Q:*** "How would the following naive baseline perform? Initialize from the expert policy, and then finetune from there with RL. It seems like this baseline could perform quite well in the proposed experimental setup, given that PPO-GAE is one of the baselines and performs quite well."
>
> ***A:*** We appreciate the suggestion! This baseline is indeed intriguing, and we have implemented and tested a similar one called MAPS-ALL (see Appendix E.1 that compares RPI against baselines with a data advantage). In MAPS-ALL, the learner policy has access to the initial pretrain transitions given by rolling-out the oracles, which is approximately twice the size accessible to the RPI algorithm. We initialize the learner policy using this extended offline data. We observe that such initialization enhances performance in the Pendulum environment (Figure 5 (c)). However, despite using significantly less data, RPI still outperforms MAPS-ALL in Walker-walk and Cartpole, and this difference is statistically significant.
>
> In addition, we followed the approach of previous works, such as MAPS and MAMBA, in using the best-so-far plot. However, we observed that MAPS-ALL often does not perform as well as expected. The primary challenge arises from "catastrophic forget" or "performance drop", which, despite not being reflected in the plot, occurs when transitioning from offline training to online fine-tuning. This challenge is a common issue in offline-to-online reinforcement learning. The prevailing method to address catastrophic forgetting involves adopting offline pessimism to online pessimism, but this often leads to suboptimal policies. Another approach is addressing catastrophic forgetting through online reinforcement learning with offline data. In contrast to the offline-to-online approach, MAPS, our RPI, demonstrates greater robustness and stability.
>
> ----
>
> We hope our response has addressed your concerns and bolstered the credibility of our work. If you have any further inquiries, please let us know. Thank you!

---

### Official Review · Reviewer_6RE6 · 2023-10-30

**Soundness:** 3 good
**Presentation:** 3 good
**Contribution:** 3 good
**Rating:** 8
**Confidence:** 3

**Summary:**

This paper investigates the setting of learning from multiple oracles that can also be framed as an imitation learning setting. The work proposes a novel procedure for combining learning from oracles with active exploration using a framework called robust active policy selection in combination with a robust policy gradient. The method interoperates between imitation learning and reinforcement learning and smoothly transitions between the two. A theoretical introduction of the presented method is followed by empirical evaluations that show improved performance over several baselines.

**Strengths:**

Motivation
* The work is well-motivated as an approach to learn from multiple oracles and improve upon them. Building frameworks that can blend between existing knowledge via oracles and exploration using a learners policy seems like a good way to enhance the setting of only learning from oracles.

Structural clarity
* The paper is well structured and well written. The flow is very clear. I do have to say that I did get lost in the notation details at times because there is a lot of notation that is quite similar. The switch from theoretical analysis to empirical investigation is handled nicely in the text.

Mathematical rigor
* The work uses strict mathematical definitions and states its assumptions clearly. The proof for proposition 4.5 is intuitive. I did not carefully check the proof of proposition 5.1 in the Appendix. However, it seems reasonable that an online no-regret learning algorithm would obtain high value.

Novelty
* The work provides a sufficiently novel version of an existing algorithm that enables a mixture of online and offline optimization. Some of the novelty might be slightly exaggerated though and could be made more clear, see Weakness: Contextualization with prior work).

Experimental evaluation
* The experiments consider a sufficient number of environments and baselines and all evaluation is done over a reasonable number of random seeds. Ablations give insight into the behavior of the algorithm and provide intuition that the algorithm works as intended.

**Weaknesses:**

Contextualization with prior work
Before I start, I would like to mention that I am not familiar with this sub-field of RL but do know the standard online and offline RL literature quite well.
* The prior work section is rather brief with a total of 6 citations. I am not familiar with the exact sub-field but this seems rather little given the literature is several years old.
* My first main concern is related to the contextualization of various parts of the paper to previous work. I think it could be made clearer which parts are inspired by previous work. For instance, the ideas of UCB value selection and multi-step advantage over several value functions seem already present in previous work but this is not immediately apparent from the current manuscript. I understand that the present manuscript defines these things with the inclusion of an additional learner policy, which requires the development of novel machinery. However, I think it would be good to give credit to the general formulations in previous work where appropriate.

Experimental evaluation
* My second main concern is the following. The experimental setting seems very similar to that studied in Liu et al., since environments and baselines are relatively similar. However, it seems that the performance of the reported baselines in this paper differs significantly from the performance reported in the previous study of MAPS. Both MAPS and MAMBA seem to be unable to even achieve performance close to the best oracle while in the MAPS paper both algorithms perform significantly better on the tasks that seem to be identical to the ones chosen here. See Q6.

Textual clarity suggestions
* This might be a me thing but the term robust policy gradient has been used several times in the literature before. It might make sense to consider a less generic name for the algorithm. Examples:

Xuezhou Zhang et al. Robust Policy Gradient against Strong Data Corruption. ICML 2021.
Yue Wang and Shaofeng Zou. Policy Gradient Method For Robust Reinforcement Learning. ICML 2022.

* I’m being a little nitpicky here but the policy gradient theorem with baseline is definitely older than 2015 (P6 section 6.2). It might be good to cite relevant prior work here.
* Some of the notation can be quite confusing and I lost track of what index refers to which object several times. This is mostly because lot’s of variables are indexed by multiple things but all objects are somewhat similar. I don’t really have a good suggestion on how to fix this though other than reducing the size of separate, less relevant sections.

Overall, I do think that the community will benefit from the publication of this work. I am inclined to raise my score if my two main concerns can be addressed.

**Questions:**

Q1. What is the relationship between the studied setting and offline RL? Offline RL, similar to the studied setting, tries to blend behavioral cloning with value function learning. In that sense, is any offline RL algorithm a candidate for learning from multiple experts? I’m not claiming that offline RL needs to be added in prior work if it turns out to be unfitting but it would be great if the authors could elaborate on this point.

Q2. Does the max-following policy execute a mixture over policies or does it, given a state, fix a policy and then execute this policy throughout time? This sentence is slightly confusing me here: “Specifically, if all oracles perform worse than the learner policy at a given state, the max-following policy will still naively imitate the best (but poor) oracle.” If the former is true, then it would be better or equal to the best poor oracle.

Q3. I’m confused about Line 5 of the algorithm. $D^k$ contains data from possibly two different policies, however it is used to update a single value function. Is this a typo or am I misunderstanding the algorithm here? I would think this should only be taking the second portion of the roll-out data, not the roll-in data. Could you elaborate why this is okay to do?

Q4. In the policy selection step, the text states “Using LCB for the learner encourages exploration unless we are very sure that the learner surpasses all oracles for the given state”. What exactly does the term exploration refer to here? Is this just meant as exploration for the learner’s policy?

Q5. To clarify that I am understanding this correctly. In Eq 14, the gradient is computed with respect to the learner’s actor policy, but with the overall multi-step advantage from the collection of value functions correct? I’m not 100% sure that I understand why this is okay since $\hat{f}^+(s_{t+1})$ and $\hat{f}^+(s_{t})$ can be from different policies, right? My guess is that it is fine because they are both independent of the action but I would appreciate some clarification.

Q6. Did you use the original implementations for MAPS and MAMBA or did you reimplement the methods? Do you have an intuition why in the present comparisons these methods perform significantly worse than in the previous literature?

---

> ### Author Response · Authors · 2023-11-17
> **Thanks for your review!**
>
> Thank you for your in-depth review of our work! We are especially grateful for your recognition of the beneficial impact that the publication of this work will have on the community. Below, please find our detailed responses to your questions.
>
>
> > ***Q1:***  "My first main concern is related to the contextualization of various parts of the paper to previous work. I think it could be made clearer which parts are inspired by previous work... I think it would be good to give credit to the general formulations in previous work where appropriate."
>
> ***A:*** Thank you for the advice! We agree that the paper would benefit from a discussion that better places RPI in the context of existing work. We created the following table in an effort to clarify the relationship between RPI and what we believe are the most relevant existing methods. Additionally, we added a detailed paragraph to Appendix A.1 that discusses related work.
>
>
>
> |         | RPI    | MAPS [ICML'23] | MAMBA [Neurips'20] |
> |--------      |--------     |--------       |  --------   |
> | Objective| Perform active IL and RL for robust policy improvement from multiple suboptimal oracles. Addressed the limitation of MAPS and MAMBA with regard to the robustness.  | Improve the sample efficiency of MAMBA by active policy selection (For MAP-SE, it also proposed active state exploration)| Perform online imitation learning from multiple suboptimal oracles. |
> | Contribution | (1) Proposed the $\text{max}^+$ framework for robust learning (2) Proposed $\text{max}^+$-aggregation baseline  (3) Robust policy improvement (RPI) algorithm (4) Robust Active Policy Selection strategy (RAPS)  (5) Robust Policy Gradient (RPG) (6) Provided theoretical analysis. | MAPS (1) Active policy selection (APS) (2) Provided theoretical analysis. | (1) Proposed max-aggregation baseline. (2) Novel algorithm MAMBA, max-aggregated baseline. (3) Regret-based performance guarantee for MAMBA.
> | Experiments |(1) Cheetah (Dense) (2) Walker-walk (Dense) (3) Cartpole (Dense) (4) Pendulum (Sparse) (5) Faucet-open (Dense) (6) Faucet-open (Sparse) (7) Drawer-close (Dense) (8) Drawer-close (Sparse) (9) Button-press (Sparse) (10)Button-press (Dense) (11) Window-close (Sparse) (12) Window-close (Dense)| (1)Cheetah (Dense)  (2)Walker-walk (Dense) (3)Cartpole (Dense) (4)Pendulum (Sparse)| (1) DIP (Dense) (2) Cheetah (Dense) (3) Ant (Dense) (4) Cartpole  (Dense)|
> | Baseline  | (1)MAPS (2)MAMBA (3)Max-aggregation (4)LOKI-variant (5)PPO-GAE (6)Best oracle| (1)MAMBA (2)PPO-GAE (3)Best oracle | (1)PG-GAE (2)Best oracle|
> | Limitation  |    lacking robustness guarantee against the optimal pure RL policy (a limitation shared by MAPS/MAMBA, although RPI demonstrates superior empirically performance).   | Not robust to the oracle quality (such as pure adversarial oracle setting)   | (1) No active oracle selection (Sample inefficiency)  (2) Not robust to the oracle quality (such as pure adversarial oracle setting) |
>
>
>
>
>
> > ***Q2:***  "My second main concern is the experimental setting seems very similar to that studied in Liu et al., since environments and baselines are relatively similar. However, it seems that the performance of the reported baselines in this paper differs significantly from the performance reported in the previous study of MAPS." "Did you use the original implementations for MAPS and MAMBA or did you reimplement the methods? Do you have an intuition why in the present comparisons these methods perform significantly worse than in the previous literature?"
>
> ***A:*** Thanks for raising this concern! We utilized the original implementation for both MAPS and MAMBA baselines. However, in our main paper, we employed a different experimental setting (specifically, the amount of accessible transitions) to ensure a fairer evaluation compared to MAPS’ setting. Additionally, we utilized a different oracle set (see footnote on Page 8). To bolster the credibility of our work, we have addressed this potential concern in Appendix E.1, where we compare RPI to baselines with a data advantage, and named MAPS as MAPS-ALL (with additional data from the oracles’ pre-trained offline dataset for learner policy.). In this section, we demonstrate that MAPS-ALL exhibits overall performance similar to that of the original MAPS paper, with any differences caused by the difference in the oracle set. Notably, as a result, MAPS-ALL distinctly outperforms RPI only in the Pendulum environment. RPI’s performance remains comparable to MAPS-ALL in the Cheetah environment and significantly surpasses the MAPS-ALL baseline in the Walker-Walk and Cartpole environments, despite utilizing much less data (~50% less).

---

> > ### Author Response · Authors · 2023-11-17
> >
> > > ***Q3:*** "The prior work section is rather brief with a total of 6 citations. I am not familiar with the exact sub-field but this seems rather little given the literature is several years old."
> >
> > ***A:*** We had originally focused the discussion on methods we felt to be most relevant to our work, while providing additional references in Appendix A "Selective Comparison Against Related Work". Following the reviewer's advice, however, we have extended the related work section with a discussion of additional methods.
> >
> >
> >
> > > ***Q4:*** "... the term robust policy gradient has been used several times in the literature before. It might make sense to consider a less generic name for the algorithm."
> >
> > ***A:*** Thanks for the advice! We are happy to use a more specific term and welcome any suggestions that you might have.
> >
> >
> > > ***Q5:*** "... policy gradient theorem with baseline is definitely older than 2015 (P6 section 6.2). It might be good to cite relevant prior work here."
> >
> > ***A:*** The reviewer is correct. Thank you for pointing this out. We have updated the text to include citations to the following papers.
> >
> > Sutton et al., Policy Gradient Methods for Reinforcement Learning with Function Approximation. JMLR, 1999
> >
> > Greensmith et al., Variance Reduction Techniques for Gradient Estimates in Reinforcement Learning. NeurIPS, 2004
> >
> >
> > > ***Q6:*** "What is the relationship between the studied setting and offline RL? Offline RL, similar to the studied setting, tries to blend behavioral cloning with value function learning. In that sense, is any offline RL algorithm a candidate for learning from multiple experts? I’m not claiming that offline RL needs to be added in prior work if it turns out to be unfitting but it would be great if the authors could elaborate on this point."
> >
> > ***A:*** Thank you for raising this question. We agree that there are similarities to our problem setting and offline RL. However, a key difference relative to offline RL is that we consider an online setting, whereby the learner is able to interact with the environment for exploration. This creates a fundamental difference from offline RL, a setting in which an algorithm must learn from (a large amount of) offline data, but is not able to interact with the environment in an online fashion. In many problems where offline RL is needed, offline RL adopts a conservative/pessimistic (safe) approach in performance action. However, in many cases, the policy is suboptimal due to dataset limitations. Online RL, in contrast, has the potential to achieve a better policy through exploration with more aggressive actions.
> >
> > > ***Q7:*** "Does the max-following policy execute a mixture over policies or does it, given a state, fix a policy and then execute this policy throughout time? This sentence is slightly confusing me here: “Specifically, if all oracles perform worse than the learner policy at a given state, the max-following policy will still naively imitate the best (but poor) oracle.” If the former is true, then it would be better or equal to the best poor oracle."
> >
> > ***A:*** The max-following policy will simply imitate the best oracle policy for any encountered state. Therefore, it will perform no worse than the single best (but still poor) oracle. However, this sentence conveys the message that for any specific state, if the learner has already outperformed the oracle, the Max-following policy will still imitate the sub-optimal oracle for that state.
> >
> >
> > > ***Q8:*** " I’m confused about Line 5 of the algorithm. $D^k$ contains data from possibly two different policies, however it is used to update a single value function. Is this a typo or am I misunderstanding the algorithm here? I would think this should only be taking the second portion of the roll-out data, not the roll-in data. Could you elaborate why this is okay to do?"
> >
> > ***A:*** Thanks for letting us know your confusion. For line 4, we first roll-in the learner policy to step $t_e$ and under state  $s_{t_e}$ in the learner’s state distribution, we roll in the best oracle to guide/correct the learner’s trajectory by using oracle policy $k$, and we then collect the second half trajectory ( the roll-out data of oracle policy) as data $D^k$. Thus, $D^k$ only contains $k_{th}$ oracle's roll out data and we use this accumulating $D^k$ to estimate the value function of oracle $k$. We have updated the pdf to make this clearer.

---

> > > ### Author Response · Authors · 2023-11-17
> > >
> > > > ***Q9:*** *"In the policy selection step, the text states “Using LCB for the learner encourages exploration unless we are very sure that the learner surpasses all oracles for the given state”. What exactly does the term exploration refer to here? Is this just meant as exploration for the learner’s policy?"
> > >
> > > ***A:*** This is a very good question. Exploration can occur in two ways. Without any Oracle policies, the learner may engage in self-exploration within a local region, but this is often less efficient, particularly for challenging tasks and sparse-reward environments. The oracle policy serves to guide the learner policy towards better states. The exploration here is the exploration guided by the oracle. We have updated the context to make it clearer.
> > >
> > > > ***Q10:*** *"To clarify that I am understanding this correctly. In Eq 14, the gradient is computed with respect to the learner’s actor policy, but with the overall multi-step advantage from the collection of value functions correct? I’m not 100% sure that I understand why this is okay since \hat{f}^{+}(s_{t+1}) and \hat{f}^{+}(s_t) can be from different policies, right? My guess is that it is fine because they are both independent of the action but I would appreciate some clarification."
> > >
> > > ***A:*** Yes, this is a good and insightful question. $\hat{f}^{+}(s_{t+1})$ and $\hat{f}^{+}(s_t)$ could be from different policies. This concept was initially introduced in Barreto et al., 2017, as Generalized Policy Improvement, where they use the $\hat{Q}$ function as $\hat{f}$ function. MAMBA extends the generalized policy improvement theorem by utilizing the Advantage function as a replacement for the original $Q$ function. By doing so, MAMBA also claims to achieve a greater policy improvement.  Intuitively, different oracles have different experiences which results in independent value functions. Compared to a single oracle or the learner policy, these additional experiences will eventually lead to an amplified advantage, increasing or decreasing the possibility of performing a specific action $a_t$ for a given transition $\{s_t,a_t.r_t,s_{t+1}\}$. As a result, this will lead to a greater policy improvement.
> > >
> > > ----
> > > We hope our response has addressed your remaining concerns, and we stay available to respond to any further inquiries you may have. Please kindly let us know. Thank you!

---

> ### Comment · Reviewer_6RE6 · 2023-11-21
>
> Thank you for the various clarifications, I appreciate it. I do think I have a better grasp of the manuscript now. I also read the other reviews as well as the responses to them.
>
> I'm coming back to a few points here to clarify what I myself was referring to.
>
> **Q1:** I was not concerned about the differences in manuscripts but rather the fact that section $4$ of the manuscript seems relatively similar in setup to the MAPS paper. In a sense, - and I'm exaggerating a little here - a large chunk of section $4$ could have been written as a summary of the MAPS paper it seems and one sentence at the end could have stated that now the learners policy is included and all functions are defined as such. The $\text{max}^+$ policy seems to be the same as the $\text{max}$ policy, $\text{max}^+$ aggregation seems to be an extension of $\text{max}$ aggregation and the $\text{max}^+$  following policy seems to be simply the definition of the $\text{max}$ following policy. In all cases, the learner's policy is added to achieve the "+".
>
> **Contextualization and Q6:** To clarify my positive novelty point. I think the general idea of adding the learners policy to the policy set is sufficiently distinct from prior approaches in the domain. This does not necessarily mean that the idea of combining offline and online training is novel. It is quite prevalent in the offline RL literature as I will outline here.
>
> I am familiar with the difference between online and offline learning. I apologize for not being more clear here. There is an abundance of literature in the offline RL community on how to leverage offline data to fine-tune and continually train a new policy by including data in replay buffers of off-policy methods [1], sampling techniques of offline and online data [2, 3, 4, 5] or simply continually fine-tuning online [6]. There is also theoretical results such as [7] on online and offline policy combination. Specifically, [5] works with the inclusion of the behavioral policy to a learning policy which seems very similar to the suggested approach. However, rather than building on a set of oracles, they use a single behavioral policy. The manuscript currently does not outline, how this work is related and how the settings are differentiated. To me it seems that they are very related which would limit both the novelty of the approach with respect to related work as well as retain the weakness of proper contextualization.
>
> [1] Vecerik et al., Deep q-learning from demonstrations. Proceedings of the AAAI Conference on Artificial Intelligence, 2018.
> [2] Nair et al., Overcoming exploration in reinforcement learning with demonstrations. In Proceedings of the International conference on robotics and automation, 2018.
> [3] Kalashnikov et al., Scalable deep reinforcement learning for vision-based robotic manipulation. In Proceedings of The 2nd Conference on Robot Learning, 2018.
> [4] Hansen et al. MoDem: Accelerating Visual Model-Based Reinforcement Learning with Demonstrations. In Proceedings of the Eleventh International Conference on Learning Representations, 2023.
> [5] Zhang et al, Policy expansion for bridging offline-to-online reinforcement learning. In Proceedings of the Eleventh International Conference on Learning Representations, 2023.
> [6] Kostrikov et al. Offline reinforcement learning with implicit Q-learning. In Proceedings of International Conference on Learning Representations, 2022.
> [7] Song et al. Hybrid RL: Using both offline and online data can make RL efficient. In Proceedings of the Eleventh International Conference on Learning Representations, 2023.
>
> As such, as before I will retain optimism in the face of uncertainty and slightly favor acceptance but I will not increase my score.
>
> * edit: formatting

---

> > ### Author Response · Authors · 2023-11-22
> > **Thank you for the detailed feedback!**
> >
> > Thank you for the detailed feedback! We appreciate you take the extra time to carefully read through our responses and reviewers, and believe that your input has greatly improve the positioning and quality of this paper. We will incorporate your suggestions in the revision; meanwhile, we hope our comments below can provide a bit more clarity on the contextualization and novelty of this work.
> >
> > > Q1: I was not concerned about the differences in manuscripts but rather the fact that section of the manuscript seems relatively similar in setup to the MAPS paper. In a sense, - and I'm exaggerating a little here - a large chunk of section could have been written as a summary of the MAPS paper it seems and one sentence at the end could have stated that now the learners policy is included and all functions are defined as such. The $\text{max}^{+}$ policy seems to be the same as the policy, $\text{max}^{+}$ aggregation seems to be an extension of aggregation and the $\text{max}^{+}$ following policy seems to be simply the definition of the max following policy. In all cases, the learner's policy is added to achieve the "+".
> >
> > ***A:*** We agree that the key difference with MAPS is the integration of the learner's policy into the (extended) policy set. However, to formally state the problem and establish our algorithm and analysis, we find it necessary to rigorously set up the notations and keep the work self-contained, which unavoiably overlaps with MAPS as a special case. In fact, many of the notations also inherit from MAMBA due to the shared problem setting, and we have moved most of the background discussion to Appendix B1 ("Additional algorithms for learning from multiple oracles") for clarity. We also add direct references to the related work to ensure that the existing works are properly acknowledged and novelty clearly stated.
> >
> > ---
> >
> > > Contextualization and Q6: To clarify my positive novelty point. I think the general idea of adding the learners policy to the policy set is sufficiently distinct from prior approaches in the domain. This does not necessarily mean that the idea of combining offline and online training is novel. It is quite prevalent in the offline RL literature as I will outline here.
> > >
> > > ... There is an abundance of literature in the offline RL community on how to leverage offline data to fine-tune and continually train a new policy by including data in replay buffers of off-policy methods [1], sampling techniques of offline and online data [2, 3, 4, 5] or simply continually fine-tuning online [6]. There is also theoretical results such as [7] on online and offline policy combination. Specifically, [5] works with the inclusion of the behavioral policy to a learning policy which seems very similar to the suggested approach. However, rather than building on a set of oracles, they use a single behavioral policy. The manuscript currently does not outline, how this work is related and how the settings are differentiated. To me it seems that they are very related which would limit both the novelty of the approach with respect to related work as well as retain the weakness of proper contextualization.
> >
> >
> > ***A:*** Thank you for clarifying your question regarding offline RL in question 6. Our initial response was based on the offline RL setting without online interaction, and we now have a clearer understanding of your inquiry. Firstly, we appreciate your inclusion of previous works, specifically those involving offline RL pretraining, followed by online RL with various sampling and fine-tuning strategies [1~7].
> >
> > The primary distinction between our work and the referenced works lies in the objectives and settings. In the RPI setting, we have access to multiple black-box oracles without an offline dataset. The technical challenge we confront involves efficiently learning from these suboptimal oracles and achieving robust learning. In contrast, the works mentioned only consider a single behavioral policy or a combination of offline and online *datasets*. This disparity in problem settings leads to substantial differences in paper objectives, theoretical analysis, and algorithm design (please see our detailed discussion below). Given this distinction, it appears that these offline-to-online RL approaches may not align well with our setting.

---

> ### Author Response · Authors · 2023-11-22
>
> > Q: Are the settings similar?
>
> ***A:*** From our perspective, our setting (the problem we aim to address) is quite different from theirs. Specifically, the approach to handling multiple suboptimal oracles and an offline dataset varies considerably. Even the treatment of multiple suboptimal oracles versus a single oracle is distinct; addressing the sample efficiency challenge to identify the state-wise optimal oracle is necessary when dealing with multiple black-box oracles, whereas a single oracle (behavioral policy) does not require tackling these challenges. To illustrate the distinction of RPI against these works, we added the referenced works [1,4~7] into Appendix A Table 2 (selective comparison against related works). Please find more details in revised version.
>
>
>
> > Q: Specifically, [5] works with the inclusion of the behavioral policy to a learning policy which seems very similar to the suggested approach. However, rather than building on a set of oracles, they use a single behavioral policy. The manuscript currently does not outline, how this work is related and how the settings are differentiated. To me it seems that they are very related which would limit both the novelty of the approach with respect to related work as well as retain the weakness of proper contextualization.
>
> ***A:*** We agree that this paper is similar to ours in that they consider an expanded policy set that includes the learner policy as well as a policy that results from offline RL. The offline policy in the expanded set is frozen and only the learner policy will be updated with online training. One of the stark differences between our work and theirs is the strategy by which we collect online data. Although we perform MAMBA-style efficient exploration that consists of rolling-in the learner policy followed by an oracle roll-out, their work relies on a rather simple mixture of predicted actions (by each policy in the set) weighted by each Q value estimate. The other difference that you have already pointed out is that they have a single pre-trained policy (i.e., a single oracle) in addition to the learner policy, whereas we consider multiple complementary oracles.
>
> ---
>
> We hope our responses have further addressed your two main concerns and clarified the remaining questions. Again, we are really appreciative of your valuable time and input!
>
> ---
>
> [1] Vecerik et al., Deep q-learning from demonstrations. Proceedings of the AAAI Conference on Artificial Intelligence, 2018.
>
> [2] Nair et al., Overcoming exploration in reinforcement learning with demonstrations. In Proceedings of the International conference on robotics and automation, 2018.
>
> [3] Kalashnikov et al., Scalable deep reinforcement learning for vision-based robotic manipulation. In Proceedings of The 2nd Conference on Robot Learning, 2018.
>
> [4] Hansen et al. MoDem: Accelerating Visual Model-Based Reinforcement Learning with Demonstrations. In Proceedings of the Eleventh International Conference on Learning Representations, 2023.
>
> [5] Zhang et al, Policy expansion for bridging offline-to-online reinforcement learning. In Proceedings of the Eleventh International Conference on Learning Representations, 2023.
>
> [6] Kostrikov et al. Offline reinforcement learning with implicit Q-learning. In Proceedings of International Conference on Learning Representations, 2022.
>
> [7] Song et al. Hybrid RL: Using both offline and online data can make RL efficient. In Proceedings of the Eleventh International Conference on Learning Representations, 2023.

---

> > ### Comment · Reviewer_6RE6 · 2023-11-22
> >
> > Thanks for your continued engagement. I thought about the setting question for a bit and I do think there are definitely differences that need to be investigated and I think this work is doing just that.
> >
> > > In fact, many of the notations also inherit from MAMBA due to the shared problem setting, and we have moved most of the background discussion to Appendix B1
> >
> > This makes sense, I think to clarify this it might be useful to move the citations to the very first paragraph in section 4 (rather than stating them under max-following), simply stating that the setting is inspired by them. This is really minor at this point but would avoid future confusions and make sure that the whole problem formulation is attributed correctly.
> >
> > I think my concerns have been addressed.
> > a.) The contextualization overall has been improved significantly.
> > b.) The experimental setup has been clarified and dissimilarities with prior work have been explained.
> > c.) My concerns with respect to similarity to offline-to-online RL have mostly been alleviated.
> >
> > Under the assumption that my previous point can be addressed quite easily, I'm happy to raise my score from 6 to 8.

---

> > > ### Author Response · Authors · 2023-11-22
> > >
> > > We are glad that our responses have addressed your previous concerns. In the revised version, we have moved the citations accordingly. We appreciate your detailed review of this paper. Additionally, we are thankful for the increase in the score and your acknowledgment of this work. Thank you!

---

### Official Review · Reviewer_yEJM · 2023-10-31

**Soundness:** 3 good
**Presentation:** 2 fair
**Contribution:** 2 fair
**Rating:** 5
**Confidence:** 3

**Summary:**

This work proposes a new policy gradient algorithm named Robust Policy Improvement (RPI), which learns policies through IL and RL. They propose the Robust Active Policy Selection (RAPS) to improve value function efficiently and then use the Robust Policy Gradient (RPG) to update policies, which is a variant of the actor-critic method. In addition to some theoretical proof of the optimality of the method, some experiments show that RPI outperforms other baselines to a degree.

**Strengths:**

1. Paper writing is clear to understand. And the method is natural.
2. The theoretical analysis is adequate.

**Weaknesses:**

(1) Novelty.
- The policy improvement of perfect knowledge is similar to making ensembles of several imitated policies. The theoretical analysis of the method seems a little redundant.
- The exploration in RPI is uniformly random sampling (line 3 of Alg.1).
These seem trivial.

(2) The experimental setting is not clear and sufficient.
- How many demonstrates for imitation learning? How many online interactions for reinforcement learning? The x-axis of the curves is training step, what is 100 training step means? And do the baselines use the same examples and online interactions?
- The selected tasks are easy. In meta-world, as I know, button-press and window-close are not hard. What about bin-picking and door-open? In DMC, what about hopper or even humanoid?
- Lack of baselines. Except for MAPS, all the baselines are from before 2020. I think there are more works than listed.

**Questions:**

As listed in weakness.

By the way, why are the links in the paper invalid? (such as citation and equations)

---

> ### Author Response · Authors · 2023-11-17
> **Thanks for your review!**
>
> Thank you for the feedback on our work! Below please find our detailed responses to your questions.
>
> ---
> > ***Q1:*** "The policy improvement of perfect knowledge is similar to making ensembles of several imitated policies... analysis seems redundant."
>
> ***A:*** We understand that our approach and ensembles of imitated policies may seem similar, but our active RL and IL strategy introduces novel technical challenges. Importantly, we do *not* assume "perfect knowledge" of the experts---unlike standard ensemble imitation, our approach requires state-wise differentiation to identify the most effective oracle in any given context (state). Moreover, a crucial aspect of our methodology is the active decision-making process, i.e., choosing between imitating the best policy or focusing on self-improvement. This aspect is particularly critical in scenarios where imitating a suboptimal policy can be detrimental, a challenge not addressed by existing ensemble-based approaches. Our learner policy is designed to determine which expert policy to follow based on the state, further differentiating our approach from traditional ensembles.
>
> ----
> > ***Q2:*** "The exploration in RPI is uniformly random sampling (line 3 of Alg.1). These seem trivial."
>
> ***A:*** Please note that we are not claiming the uniform sampling of the roll-in horizon $t_e$ in Line 3 of Alg. 1 as novelty of this work. Rather, we focus on the robust *policy selection* and robust *policy gradient* algorithm, which ensures active exploration of RPI.
>
> Selecting an oracle to imitate uniformly at random from the learner’s state distribution is a standard procedure in online imitation learning. As the reviewer correctly points out, this strategy, employed in several classic online IL algorithms like Dagger (Ross et al., 2011), Aggravate (Sun et al., 2017), THOR (Sun et al.,2018), MAMBA (Cheng et al.,2020), and MAPS (Liu et al.,2023), serves as a common practice for conducting imitation learning based on the learner's distribution, ensuring comparability with existing methodologies.
>
> > ***Q3:*** "The experimental setting is not clear and sufficient. How many demonstrates for imitation learning? How many online interactions for reinforcement learning? The x-axis of the curves is the training step, what does 100 training steps mean? And do the baselines use the same examples and online interactions?"
>
> ***A:*** Thank you for raising these questions. We would like to note that most of these experimental details were presented in the appendix (D.3 setup D.4 implementation details and D.5 hyperparameter table) and in the source code---both included in the supplemental materials---for reproducibility of our results. In terms of your specific questions:
> * We will explain the details of each training iteration as follows. For each training iteration of the RPI algorithm, the environment is queried for 4 episodes (roll-in + roll-out; this amounts to 4k steps for DMControl suite, 1.2k steps for MetaWorld tasks), followed by 2,048 steps (only rolling out the learner's policy). Thus, each training iteration involves 6,048 steps of environment interactions for DMControl suite (or 3,248 steps of interactions for MetaWorld).
> * To elaborate on the learner's policy update within a single training iteration, we will give a little bit of implementation details: 1. We first prepare a set of transitions. This comes from the roll-in (see Alg. 1 line 4) and full roll-out (see Alg. 1 line 6) of the learner policy. 2. We then create a set of mini-batches (with size 128) by grouping the transitions. 3. We iteratively perform gradient updates on the mini-batches for 2 epochs (i.e., going over all of the mini-batches twice). This entire process makes up a single training step.
> * All baselines were subjected to an equal number of environment interaction steps to maintain consistency, except for a minor difference in PPO-GAE (please see Appendix D.3 Setup).
>
> We have updated the text in the main paper and appendix to make sure that the experimental details are clear.

---

> ### Author Response · Authors · 2023-11-17
>
> > ***Q4:*** " The selected tasks are easy. In the meta-world, as I know, button-press and window-close are not hard. What about bin-picking and door-open? In DMC, what about hopper or even humanoid?"
>
> ***A:*** Please note that the tasks considered were variants of the standard setting, tailored towards the evaluation of imitation learning with multiple blackbox oracles:
>
> * Button-press and window-close are originally dense-reward environments (see Appendix E.2 Meta-world experiments (dense reward)). However, we converted the environment to have a sparse reward, where a reward is given only when the window is closed or a button is pressed. Thus, most of the time, the learner doesn't receive any feedback. Compared to the original dense reward environment, this is significantly harder learning problem.
>
> * We chose these tasks to facilitate comparisons with the most relevant work in online imitation learning as they are a superset of the standard baselines for this domain. Please refer to the following table, which compares tasks across methods, and kindly check our updated PDF for more details.
>
>
> |         | RPI    | MAPS [ICML'23] | MAMBA [Neurips'20] |
> |--------      |--------     |--------       |  --------   |
> | Experiments |(1) Cheetah (Dense) (2) Walker-walk (Dense) (3) Cartpole (Dense) (4) Pendulum (Sparse) (5) Faucet-open (Dense) (6) Faucet-open (Sparse) (7) Drawer-close (Dense) (8) Drawer-close (Sparse) (9) Button-press (Sparse) (10)Button-press (Dense) (11) Window-close (Sparse) (12) Window-close (Dense)| (1)Cheetah (Dense)  (2)Walker-walk (Dense) (3)Cartpole (Dense) (4)Pendulum (Sparse)| (1) DIP (Dense) (2) Cheetah (Dense) (3) Ant (Dense) (4) Cartpole  (Dense)|
>
>
>
> > ***Q5:*** " Lack of baselines. Except for MAPS, all the baselines are from before 2020. I think there are more works than listed."
>
> ***A:*** To the best of our knowledge, there are only a few existing approaches that focus on online imitation learining from multiple black-box oracles. We included the state-of-the-art MAPS [2023], along with all baselines used to evaluate MAPS, such as MAMBA [2020]. Additionally, we implemented several variants of the proposed algorithm to provide a comprehensive evaluation. To provide some further context, please refer to `Appendix Table 2` for a detailed comparison to 17 other relevant methods, which explicitly points out the key differences in problem settings.
>
> We believe we have included a sufficient set of relevant baselines for the problem setting that we consider, but would appreciate the reviewer's suggestion for any other relevant baselines.
>
>
> |         | RPI    | MAPS [ICML'23] | MAMBA [Neurips'20] |
> |--------      |--------     |--------       |  --------   |
> | Baseline  | (1)MAPS (2)MAMBA (3)Max-aggregation (AggreVaTed-variant for muliple oracles) (4)LOKI-variant (5)PPO-GAE (6)Best oracle| (1)MAMBA (2)PPO-GAE (3)Best oracle | (1)PG-GAE (2)Best oracle (AggreVatD(single oracle setting))|
>
>
> ----
> > ***Q6:*** "... why are the links in the paper invalid? (such as citation and equations)"
>
>
> ***A:*** We appologize for the inconvenience. It was a result of generating the PDF of the main paper by removing the appendix pages from a version of the paper that included the main text and appendices as a single PDF. Please see a complete version of the paper (including the appendices) with working links in the supplementary materials section. We note that many of the questions that you raised regarding environment setup have been addressed there as well.
>
>
> ----
>
> We hope these responses adequately address your concerns and clarify the contributions and methodologies of our work. We appreciate your constructive feedback and look forward to further discussions!

---

> ### Comment · Reviewer_yEJM · 2023-11-21
> **Official Comment by Reviewer yEJM**
>
> Thank you for the response. The experiment setup is more clear and sufficient. But I think the novelty is quite limited. I have raised my score to 5.

---

> > ### Author Response · Authors · 2023-11-21
> > **Thank you for the additional feedback!**
> >
> > Thank you for the additional feedback! We are glad that our initial response helped clarify the sufficiency of our empirical study. We would like to take the opportunity to further clarify the novelty of this work.
> >
> > ---
> > > "...the novelty is qutie limited..."
> >
> > |         | RPI    | MAPS [ICML'23] | MAMBA [Neurips'20] |
> > |--------      |--------     |--------       |  --------   |
> > | Objective| Perform active IL and RL for robust policy improvement from multiple suboptimal oracles. Addressed the limitation of MAPS and MAMBA with regard to the robustness.  | Improve the sample efficiency of MAMBA by active policy selection (For MAP-SE, it also proposed active state exploration)| Perform online imitation learning from multiple suboptimal oracles. |
> > | Contribution (novelty) | (1) Proposed the $\text{max}^+$ framework for robust learning (2) Proposed $\text{max}^+$-aggregation baseline  (3) Robust policy improvement (RPI) algorithm (4) Robust Active Policy Selection strategy (RAPS)  (5) Robust Policy Gradient (RPG) (6) Provided theoretical analysis. | MAPS (1) Active policy selection (APS) (2) Provided theoretical analysis. | (1) Proposed max-aggregation baseline. (2) Novel algorithm MAMBA, max-aggregated baseline. (3) Regret-based performance guarantee for MAMBA.
> >
> > Besides the above distinction between RPI and previous works, we also want to highlight some technical challenges in developing the RPI algorithm.
> >
> > ***1. Theoretical analysis***
> >
> > The previous approach MAMBA[Neurips'20]/MAPS[ICML'23] all used a static oracle set. This theoretical analysis doesn't apply to a novel, dynamically extended oracle setting. To address this theoretical analysis challenge, we conducted nontrivial theoretical analysis to transition the setting from a static oracle setting to a dynamic oracle setting. Additionally, we also proved that our approach improves the previous performance lower bound in the state-of-the-art (SOTA), MAMBA.
> >
> > ***2. Experiment design.***
> >
> > To conduct a more robust evaluation, we performed a more comprehensive assessment in both sparse and dense environments. In addition to the standard Meta-world benchmarks on Window-close, Faucet-open, Drawer-close, Button-press, which initially feature dense rewards, we also contructing more challenging variants of these environments with spare rewards. These variants amount to a total collection of 12 environments with a mix of spare and dense conditions, leading to a robust evaluation of RPI.
> >
> >
> > ***3. Algorithm design in RPI.***
> >
> > In RPI, we first propose Robust Active Policy Selection (RAPS). MAPS [ICML'23] introduced active policy selection. However, we found that in the case when the learner has outperformed the oracles, there is no need to roll out the sub-optimal oracle on those states, especially in the later stage. Different from MAPS which relies on a pure UCB strategy based on the oracle set, we propose to use LCB of the learner's value function and the UCB of the oracle set. This facilitates self-improvement and, at the same time, encourages oracle-guided exploration unless we are certain that the learner surpasses all oracles for the given state. After incorporating the adaptive learner into the oracle set, we found that the previous actor-critic framework doesn't fit for a dynamic oracle set. Thus, we introduce the $\text{max}^{+}$ actor-critic framework, where the actor samples trajectories that are then evaluated by the $\text{max}^{+}$ critic based on the $A^{GAE+}$ advantage function, which enables the learner policy $\pi_n$ to learn from high-performing oralce and improve its own value function $\hat{V}^k$ for the states in which the oracles perform poorly.
> >
> > ---
> > We hope our response could further clarify the novelty of this work. Please kindly let us know if there are any further questions. Thank you!

---

### Meta-Review · Area_Chair_KNzB · 2023-12-06

**Metareview:**

This paper proposes a method for combining reinforcement learning with imitation learning and demonstrates its effectiveness on a wide range of established benchmark tasks. The majority of reviewers noted significant value in the paper's contribution, with the one critical reviewer not fully justifying a claimed lack of novelty. All reviewers have offered suggestions for improvements to the paper, many of which the authors have already acted on. The paper is likely to be of broad interest to researchers in the decision making sub-community within ICLR.

**Justification For Why Not Higher Score:**

Whilst weakly justified by Reviewer yEJM, there are some remaining doubts on the novelty of the method which may limit its appeal beyond researchers interested in sequential decision making.

**Justification For Why Not Lower Score:**

Broad appeal across two subcommunities in decision making (imitation learning and reinforcement learning)

---

### Decision · Program_Chairs · 2024-01-16

Accept (spotlight)